

# Classifying irreducible fixed points of five scalar fields in perturbation theory

**Junchen Rong⋆ and Slava Rychkov†**

Institut des Hautes Études Scientifiques, 91440 Bures-sur-Yvette, France

⋆ junchenrong@ihes.fr , † slava@ihes.fr

## Abstract

Classifying perturbative fixed points near upper critical dimensions plays an important role in understanding the space of conformal field theories and critical phases of matter. In this work, we consider perturbative fixed points of $N = 5$ scalar bosons coupled with quartic interactions preserving an arbitrary subgroup $G \subset O(5)$. We perform an exhaustive algorithmic search over the symmetry groups $G$ which are irreducible and satisfy the Landau condition, so that the fixed point can be reached by fine-tuning a single mass term and there is no need to tune the cubic couplings. We also impose stability of the RG flow in the space of quartic couplings, and reality. We thus prove that there exist no new stable fixed points in $d = 4-\epsilon$ dimensions beyond the two known ones: namely the O(5) invariant fixed point and the Cubic(5) fixed point. This work is a continuation of the classification of such fixed points with $N = 4$ scalars by Toledano, Michel, Toledano and Brézin in 1985 [1].



# 1 Introduction

Classification of critical states of matter is an important problem, which can be approached from many different directions. One possibility is to classify renormalization group (RG) fixed points in perturbation theory close to the upper critical dimension. This approach provides interesting information, since found fixed points often continue down to the physical dimensions.

In this paper we consider RG fixed points for the theory of $N$ scalar fields in $d = 4 - \varepsilon$ dimensions, described by the Lagrangian

$$\frac{1}{2}(\partial_\mu \phi^i)^2 + \frac{1}{2}\mu_{ij}\phi^i\phi^j + \frac{1}{3!}c_{ijk}\phi^i\phi^j\phi^k + \frac{1}{4!}\lambda_{ijkl}\phi^i\phi^j\phi^k\phi^l. \tag{1}$$

At the fixed point, the mass term $\mu_{ij}$ and the cubic couplings $c_{ijk}$ have to be tuned to zero, while the tensor of quartic couplings, assumed real, has to satisfy the beta function equation

$$\beta_{ijkl} = -\epsilon\lambda_{ijkl} + \frac{1}{16\pi^2}\left(\lambda_{ijmn}\lambda_{mnkl} + 2 \text{ permutations}\right) + O(\lambda^3). \tag{2}$$

The most important characteristic of the fixed point is its global symmetry group $G$, which is defined as the maximal subgroup of $O(N)$ leaving (1) invariant. Excluding the Gaussian fixed point $\lambda = 0$, the most symmetric fixed point is the Wilson-Fisher fixed point [2] which preserves the full $O(N)$. For other symmetries, the most interesting case arises when the group $G$ is large enough so that three conditions are satisfied:

I. (Irreducibility) There is a single mass term compatible with $G$, namely the fully isotropic one $\mu_{ij} = \mu\delta_{ij}$;

L. (Landau condition) No cubic term is allowed by symmetry $G$;

S. (Stability) All quartic couplings allowed by symmetry $G$ are irrelevant, in the RG sense, at the fixed point.

If these conditions are satisfied, the fixed point can describe a continuous phase transition (critical point) obtained by tuning a single relevant coupling consistent with the symmetry (the

isotropic mass). Violating any of these conditions would imply multiple tunings necessary to reach the fixed point. This would correspond to multi-critical behavior, making it more difficult to realize such fixed points experimentally.

The irreducibility condition is called so because mathematically, a single mass term means that the vector representation of O($N$) remains irreducible over the real numbers under $G$.[1] Such subgroups of $G$ are called irreducible. The Landau condition is important because phase transitions violating this condition are expected to be first order, unless of course extra tuning is performed to tune the cubic couplings to zero.[2] Stability condition is similarly motivated - phase transition will be first order if the microscopic theory does not belong to the basin of attraction of a stable fixed point, in particular, if no stable fixed point exists.

Note that conditions I and L depend only on the symmetry group $G$, while verifying condition S requires solving the beta function equations. Fixed points satisfying all three conditions will be called ILS fixed points. It is interesting to classify, for each $N$, such fixed points.

For $N = 2, 3$ the only ILS fixed point is the O($N$) fixed point. For $N = 4$ there are four ILS fixed points which were classified many years ago by Toledano, Michel, Toledano and Brézin [1], using the mathematical knowledge of finite subgroups of O(4). It turns out that classification of finite subgroups of O(5) is also available [3]. Using this result, in this paper we will classify ILS fixed points for $N = 5$.

We will start in Section 2 by describing the classification of [1] as well as the results available for other $N$. Then in Section 3, we introduce the general strategy to study general Landau theories with $N$ scalars. Certain group theory notions are introduced for later convenience. In Section 4, we discuss the maximal irreducible subgroups of O(5), drawing on various mathematical results. In Section 5 we present an algorithm which generates all the irreducible subgroups of O($N$) and their group-subgroup relations starting from a list that is known to contain all the maximal irreducible subgroups of O($N$). The results of applying this algorithm for $N = 5$ are summarized in Section 6. These results allow us to greatly reduce the number of Landau theories we need to consider. We describe these Landau theories in Section 7 and then study their RG flows in Section 8. Our results and future directions are discussed in Section 9. Group theory computations of this paper are done using GAP [4].

## 2 Review of known results

In this section we review what is known about ILS fixed points for various $N$. All statements about the existence and stability of fixed points are at one loop unless otherwise specified.

The O($N$) fixed point is stable with respect to O($N$)-symmetric perturbations for any $N$. In $N = 2, 3$ the O($N$) fixed point is globally stable, while any other fixed point is unstable [5]. This is in accord with a theorem of Michel [6] (see [7] for a review): at one-loop order, for any symmetry $G$ the number of stable fixed points preserving symmetry $G$ or larger is at most one.

*Remark* 2.1. In this paper, all statements about fixed point stability refer to $d = 4 - \varepsilon$ dimensions, unless stated otherwise. It does sometime happen that fixed points which are stable near 4 dimensions become unstable in 3 dimensions, and vice versa. This may happen because two fixed points collide in some intermediate dimension and exchange their stability properties. Here are two famous examples of this phenomenon:

---

[1]An $N$-dimensional representation irreducible over the reals maybe be reducible over complex numbers. It then becomes a pair of complex conjugate irreps with dimension $N/2$. Clearly, this may happen only if $N$ is even. In this paper, we focus on $N = 5$ so for us this never happens. Unless otherwise specified, "irreducible" in our paper stands for irreducible over the real numbers.

[2]In low dimensions, the Landau condition may be relaxed because cubic couplings may become irrelevant, as happens famously for the 3-state Potts model in two dimensions.

- In $N = 3$, the O(3) fixed point is stable and the cubic fixed point, see below, is unstable in $d = 4 - \varepsilon$, but the stability properties are reversed in $d = 3$ [8–10].

- In $N = 4$, the fixed point of two decoupled O(2) models is unstable, while the O(2)×O(2) fixed point, see below, is (two-loop) stable. In $d = 3$ the stability is reversed.

For $N = 4$, the O(4) fixed point is, at one loop, marginal with respect to O(4) breaking perturbations [5]. This implies that any fixed point differing from O(4) at one loop is unstable [5]. To resolve the degeneracy of O(4) breaking directions requires two-loop analysis [1], which we discuss below.

For $N \geqslant 5$ the O($N$) fixed point is one-loop unstable in O($N$) breaking directions, which opens the possibility to having one-loop stable fixed points with symmetry smaller than O($N$).

Before discussing full classification results in $N = 4$ [1] and in $N = 5$ (this work), as well as partial results available for $N = 6$ [11–13], let us mention some infinite series of fixed points.

There are several infinite families of I&L symmetries $G$ which have only two quartic invariants, so that the quartic Landau theory contains only two couplings. In this case there are three fixed points: O($N$) and two $G$-invariant ones. For $N \geqslant 5$, one of the two $G$-invariant fixed points is stable, the other unstable [5, 14] (also reviewed in [7]). Two-loop analysis is needed if these two fixed points are one-loop degenerate. Symmetries falling into this category are:

- cubic: Cubic($N$) = $(\mathbb{Z}_2)^N \rtimes S_N$ [15]. There is a one-loop stable cubic fixed point for all $N \geqslant 5$. For $N = 4$ it is degenerate with the O(4) fixed point, but becomes distinct from it, and stable, from two loops on [1]. For $N = 3$ the cubic fixed point is one-loop unstable (see however Remark 2.1).

- tetrahedral: $S_{N+1} \times \mathbb{Z}_2$ [14, 16] (also known as "restricted Potts model"). There is a one-loop stable fixed point for all $N \geqslant 6$. For $N = 4$ it is stable, after resolving degeneracy from O(4) at two loops [1]. At $N = 5$ the fixed point has complex couplings (at two loops), see Section 8.3.

- orthogonal bifundamental: O($p$) × O($q$)/$\mathbb{Z}_2$, $N = pq$. There is a real fixed point at one loop if [14, 17]
$$R_{pq} = p^2 + q^2 - 10pq - 4(p + q) + 52 \geqslant 0. \tag{3}$$

    Restricting to $p \geqslant q$ this condition is satisfied for $p = q = 2$ and for $p \geqslant 5q + 2 + 2\sqrt{6(q-1)(q+2)}$ which requires large values of $p$ e.g. $p \geqslant 22$ for $q = 2$. The $p = q = 2$ fixed point coincides with the O(4) fixed point. It becomes distinct, and stable, at two loops [1] (see however Remark 2.1).

- unitary bifundamental: U(p) × U(q)/U(1) [18–20]. The Lagrangian is written most naturally in terms of a $p \times q$ complex matrix field. In terms of real components we have $N = 2pq$ The corresponding Lagrangian is different from the O($N$) Lagrangian when $p \geqslant 2$ and $q \geqslant 2$, so that $N \geqslant 8$. For a fixed $p$, a pair of new fixed points exist when $q > q^+$ and $q < q^-$. One of these two fixed points is stable when $q < q^-$ and $q > q^+$. At one loop order, we have $q^\pm = 5p \pm \sqrt{6(p^2 - 1)}$.

- "MN": $\Gamma_{p,q} = O(p)^q \rtimes S_q$, $N = pq$. [14, 21]. The case $p = 1$ is equivalent to the cubic, and $p = q = 2$ to the bifundamental. There are two fixed points, one fully interacting and another factorized ($q$ copies of O($p$) fixed points). The factorized fixed point is one-loop stable for $p > 5$. For $p = 4$ the two fixed points coincide at one loop so they are marginally stable. For $p = 2, 3$ the fully interacting fixed point is stable.

There are very few examples of stable fixed points having symmetry groups with 3 or more quartic invariants. A series of examples was constructed in [22] (see also [14,23]). His smallest example has $N = 12$, $G = \Gamma_{2,6} \cap \Gamma_{6,2}$, and three quartic invariants.

Finally, let us discuss classifications for $N = 4$ and $N = 5$ and partial results for $N = 6$.

In $N = 4$, Toledano et al [1] classified all I&L subgroups of O(4), their quartic Landau theories and the fixed points. Using two-loop analysis when needed, they proved that there exist exactly four ILS fixed points, having symmetries O(4), Cubic(4), $S_{N+1} \times \mathbb{Z}_2$, $O(2)^2 \rtimes \mathbb{Z}_2$. All of them are members of the above families.[3]

In this work, we will perform a similar full classification in $N = 5$. Our result is that, apart from O(5) and Cubic(5), there are no new stable fixed points.

In $N = 6$, a partial classification was carried out in [11–13]. These authors consider subgroups of $O(6)$ which can be realized as images of 6-dimensional representations of one of 230 space groups in three dimensions. Their work is thus relevant to the study of second-order structural phase transitions. They identify 11 such images, which are irreducible subgroups of $O(6)$. It's a large family of subgroups, but the only stable fixed point they identified was of the "MN" type, namely $O(2)^3 \rtimes S_3$.

## 3 General strategy

Given any symmetry group $G \subset O(N)$, it is in principle straightforward to classify ILS fixed points having this symmetry. First one checks whether conditions I and L are satisfied. These condition say that $I_2 = 1$, $I_3 = 0$, where $I_2$ and $I_3$ are the numbers of linearly independent quadratic and cubic invariant polynomials. They may be computed e.g. using the Molien function. (See Eq. (23) for more details.) To verify the condition S, one first needs to find the basis of quartic invariant polynomials, call it $P_{4,a}(\phi^i)$, $a = 1, \ldots, I_4$. Then one writes down the quartic Landau theory consistent with the symmetry (setting the quadratic term to zero)

$$L(\phi) = \frac{1}{2}(\partial_\mu \phi^i)^2 + \sum_{a=1}^{I_4} \lambda_a P_{4,a}(\phi^i), \tag{4}$$

computes beta functions for the quartic couplings $\lambda_a$, finds fixed points, and selects those of them, if any, which are real and RG stable.

In general, the group $O(N)$ has infinitely many subgroups. Our goal is to find all ILS fixed points. One way to approach this goal would be to first find all subgroups satisfying conditions I and L, and then verify condition S for each of them as described above. This is still a bit tedious since the number of I&L subgroups is significant. In this paper we will speed up the procedure further by using group-subgroup relations. Our procedure will consist of three steps:

1. Generate a list containing all maximal irreducible subgroups of O(N). In our case ($N = 5$), this will be done with the help of mathematical literature classifying finite subgroups of O(5).

2. Given the list from Step 1, generate all irreducible subgroups of $O(N)$, and their group-subgroup relations. In practice, we have an algorithm which deals with finite irreducible subgroups. The irreducible subgroups which are Lie groups have to be dealt with separately. For the $N = 5$ case this turns out to be easy, since there are very few of them.

3. Study the Landau theories of the *minimal I&L* subgroups, to find their fixed points and select the stable ones. Here "minimal I&L" means not having any subgroups satisfying I&L conditions.

---

[3]In [1], the tetrahedral symmetry is called di-icosahedral, and $O(2)^2 \rtimes \mathbb{Z}_2$ bi-cylindrical.

Step 3 is the crucial simplification - instead of having to study many Landau theories, we will have to study only a few. Of course this is only possible if one knows which subgroups are minimal - something we will know thanks to Step 2.

## 3.1  Matrix groups vs abstract groups

One subtlety of the discussion below is that we will have to talk about two different kinds of groups. The most important for us are symmetry groups of Landau theories, which are matrix groups, subgroups of $O(N)$. A matrix group is a set of matrices with group multiplication given by matrix multiplication.[4] Two matrix groups $G$ and $G'$ are considered equivalent for our purposes (written $G \cong G'$) if they are conjugate within $O(N)$, i.e. there is a similarity transformation $g \to SgS^{-1}$, where $S \in O(N)$ is a fixed matrix, which maps the set of matrices forming $G$ onto the matrices forming $G'$. We will use "conjugate within $O(N)$" and "equivalent" interchangeably.

We will also have to talk about abstract groups, i.e. about sets with a multiplication operation, not necessarily realized as matrices. To avoid confusion, we use Latin letters $G, H, \dots$ to denote matrix groups, and Gothic letters $\mathfrak{G}, \mathfrak{H}, \dots$ to denote abstract groups.

Two abstract groups $\mathfrak{G}$ and $\mathfrak{G}'$ are called isomorphic if there is a one-to-one map from one to the other which preserves the multiplication rule.

Every matrix group $G$ gives rise to the corresponding abstract group $\mathfrak{G}$ which is obtained from $G$ by extracting its multiplication table. Obviously, if $G$ and $G'$ are conjugate within $O(N)$, then the corresponding abstract groups $\mathfrak{G}$ and $\mathfrak{G}'$ are isomorphic. But the converse is not necessarily true: it may happen that two matrix groups $G_1$ and $G_2$, while not conjugate to each other, are both isomorphic to the same abstract group $\mathfrak{G}$. Such matrix groups will be considered as different groups for our purposes, and will be denoted accordingly.

We want to discuss the opposite operation: how given an abstract group $\mathfrak{G}$ to obtain all inequivalent matrix groups $G \subset O(N)$ isomorphic to $\mathfrak{G}$.

Clearly, we can obtain a matrix group $G \subset O(N)$ from an abstract group $\mathfrak{G}$ by using any real $N$-dimensional representation $\rho$ of $\mathfrak{G}$, and defining $G$ as a set of matrices representing the elements of $\mathfrak{G}$ in this representation. In addition, if the representation is faithful, the so obtained matrix group $G$ will be isomorphic to $\mathfrak{G}$ as an abstract group. The faithfulness condition is important: if it is violated, then some elements of $\mathfrak{G}$ will be mapped to identical matrices. The set of non-identical matrices obtained this way will still form a matrix group, but it will not be isomorphic to $\mathfrak{G}$.

If two representations $\rho_1$ and $\rho_2$ are isomorphic in the sense of representation theory, then the two matrix groups $G_1$ and $G_2$ obtained this way will clearly be conjugate within $O(N)$.

There is however another subtlety. Sometimes two matrix groups $G_1$ and $G_2$ will be conjugate within $O(N)$ even if $\rho_1$ and $\rho_2$ are not isomorphic representations. Such cases are related to outer automorphisms of the abstract group $\mathfrak{G}$.

An automorphism is an isomorphism from a group to itself, which respects the multiplication table of the group. The group of automorphisms, $\text{Aut}(\mathfrak{G})$, acts on representations of the group. If $\rho : \mathfrak{G} \to \text{End}(V)$ is a representation, and $f : \mathfrak{G} \to \mathfrak{G}$ is an automorphism, then $\rho^f = \rho \circ f$ is another representation. Automorphisms can be inner and outer. Inner automorphisms, forming the group $\text{Inn}(\mathfrak{G})$, are (abstract) conjugations

$$g \to h^{-1}gh, \qquad h \in \mathfrak{G}. \tag{5}$$

For inner automorphisms, representation $\rho^f$ is isomorphic to $\rho$. Outer automorphism group of group $\mathfrak{G}$ is defined as $\text{Aut}(\mathfrak{G})/\text{Inn}(\mathfrak{G})$. For outer automorphisms, $\rho^f$ may not be isomorphic to

---

[4]Notice that since we use "set" and not "list", all matrices are distinct.

$\rho$. However, the matrix groups constructed from $\mathfrak{G}$ using $\rho$ and $\rho^f$ clearly will be equivalent, since they will consist of the same matrices, up to reshuffling.

The bottomline of this discussion is the following basic fact.

**Lemma 3.1.** *We will obtain all inequivalent matrix groups $G \subset O(N)$ from an abstract group $\mathfrak{G}$ by using all real N-dimensional faithful representation $\rho$ of $\mathfrak{G}$, up to equivalence by outer automorphisms.*

To apply this criterion we have to know when two group irreps are equivalent by an outer automorphism. The crucial property of an outer automorphism is that it acts as a permutation on conjugacy classes of the group, and on characters of representations of the group. We will have to identify permutations corresponding to all outer automorphisms, and identify characters related by these permutations.

*Example* 3.1. The abstract group $S_6$ has 4 different faithful 5-dimensional irreps, but only two of them are non-equivalent, up to outer automorphisms. This means that there are two different matrix groups $G_1, G_2 \subset O(5)$ which are both isomorphic to $S_6$ as abstract groups, which we will denote as $S_6$–$A$ and $S_6$–$B$.

Let us next discuss group-subgroup relations. If $G_1 \subset O(N)$ and $G_2 \subset O(N)$ are two matrix groups, we say that $G_1 \subset G_2$ if $G_1$ is conjugate to a subgroup of $G_2$. Note that if $G_1 \subset G_2$ in this sense, then we also have $\mathfrak{G}_1 \subset \mathfrak{G}_2$ at the abstract group level, where $\mathfrak{G}_1, \mathfrak{G}_2$ are the corresponding abstract groups. But the converse is not necessarily true: if we take two matrix groups $G_1$ and $G_2$ constructed from $\mathfrak{G}_1, \rho_1$ and $\mathfrak{G}_2, \rho_2$, with $\mathfrak{G}_1 \subset \mathfrak{G}_2$, then not necessarily $G_1 \subset G_2$ as matrix groups.

## 3.2 Steps 2&3

With definitions out of the way, let us give more details about Steps 2&3 of the above strategy. Below we will present an algorithm to generate the list of irreducible subgroups of $O(N)$ from a list containing all maximal irreducible subgroups, and simultaneously figure out group-subgroup relations among all of them. Applying this algorithm, we will organize the irreducible subgroups in a directed graph, schematically shown in Fig. 1, where on each level we put subgroups of the groups from the previous level. Furthermore, for each subgroup in the graph, we can check whether the Landau condition is satisfied or not (denoted by a star). Clearly, if this condition holds for any subgroup $\mathcal{G}$, then it holds for all larger subgroups $G'$, $G \subset G' \subset O(N)$.

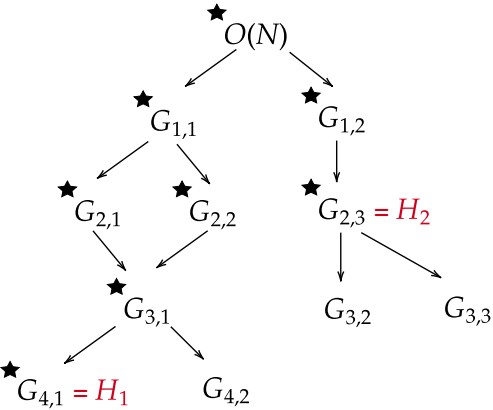

Figure 1: Graph encoding group-subgroup relations, see the text.

Given such a graph, we consider *minimal* I&L subgroups $H_1, \ldots, H_r$ ($r = 2$ in Fig.1). We will study Landau theories only for these minimal I&L subgroups. Indeed, when we study the RG flow of these Landau theories and search for fixed points, we are guaranteed to find all the fixed points whose symmetry is larger or equal to $H_i$, which by definition includes all I&L subgroups.

Another feature of our analysis will be the phenomenon of symmetry enhancement, discussed in Section 7.2.1 below.

# 4   Maximal irreducible subgroups of O(5)

We will now carry out the above strategy for $N = 5$. The first step is to make a list containing all maximal irreducible subgroups of O(5). We first discuss finite subgroups of O(5), and then Lie subgroups of O(5).

## 4.1   Finite subgroups of O(5)

The finite subgroups $G \subset$ O(5) were classified in [3] where they were shown to fall into the following three classes

- $G$ is (conjugate to) a subgroup of $(\mathbb{Z}_2)^5 \rtimes S_5$, called Cubic(5).

- $G$ is, as an abstract group, isomorphic to $A_5, S_5, A_6$ or $S_6$, or to the direct product of one of them with $\mathbb{Z}_2$, where, in the matrix group realization, the $\mathbb{Z}_2$ factor is represented as

$$\mathbb{Z}_2^{O(5)} = \{I, -I\}, \tag{6}$$

  where $I$ is the $5 \times 5$ identity matrix.

- $G$ is (conjugate to) a subgroup of O(3) $\times$ O(2) or O(4) $\times \mathbb{Z}_2$.

In the last class, the O(3) $\times$ O(2) or O(4) $\times \mathbb{Z}_2$ subgroups are embedded in O(5) in the natural way, or in other words, the 5 dimensional vector irrep of O(5) decomposes as $3 + 2$ and 4+1. This means that none of these subgroups are irreducible, and we can ignore them.[5]

This mathematical result gives us one matrix group and four abstract groups:

$$\text{Cubic(5)} \qquad \text{(matrix group)}$$
$$A_5 \times \mathbb{Z}_2, S_5 \times \mathbb{Z}_2, A_6 \times \mathbb{Z}_2, S_6 \times \mathbb{Z}_2 \qquad \text{(abstract groups)} . \tag{7}$$

Maximal irreducible subgroups of O(5) should be Cubic(5) or isomorphic, as abstract groups, to one or more of the abstract groups in Eq. (7), although at this stage we can't yet say to which ones.

As a next step we have to find matrix groups corresponding to the abstract groups in (7). By Lemma 3.1, this amounts to finding their real 5-dimensional faithful representations which are irreducible over real numbers, irreducible since we are interested in irreducible subgroups, and up to outer automorphism equivalence.

Let us discuss how this is done for the abstract group $S_5 \times \mathbb{Z}_2$, other cases being similar. All group theory calculations are done using GAP [4], see App. C. In GAP, we specify $S_5 \times \mathbb{Z}_2$ group by

```
g:=DirectProduct(SymmetricGroup(5),CyclicGroup(2));
```
(8)

---

[5]As a side comment, we note that this class of subgroups can be further specified with the help of the knowledge of finite subgroups of $O(4)$ [24] and O(3) [25]. The finite subgroups of O(3) famously follow an ADE classification.

Once the group is specified, we compute its character table:

$$\texttt{tbl:=CharacterTable(g); Display(tbl);} \tag{9}$$

and look up the faithful 5-dimensional irreps. Recall that the dimension $n$ of the representation equals to $\chi(e)$, character at the identity. A simple criterion for detecting unfaithful representations is this:

**Lemma 4.1** ( [26], p.5). *An n-dimensional representations is unfaithful if and only if its character $\chi$ takes value n more than once.*

These representation are not necessarily real representations. To select the real representations, we can calculate their (second) Frobenius–Schur indicators by

$$\texttt{Indicator(tbl,2);} \tag{10}$$

which equals to 1, 0 and $-1$ for real, complex and quaternionic representations respectively. Here `tbl` is the character table calculated in (9).[6]

For $S_5 \times \mathbb{Z}_2$, we find that there are 2 non-isomorphic faithful 5-dim irreps, which are the 11th and 12th irreps of the group. We next have to check whether these irreps are equivalent by an outer automorphism. Automorphism groups can be easily constructed in GAP:[7]

```
AutG:=AutomorphismGroup(g);
InnG:=InnerAutomorphismsAutomorphismGroup(AutG);
outerautomorphisms:=List(RightCosets(AutG,InnG),c->Representative(c));
```
$$\tag{11}$$

For $S_5 \times \mathbb{Z}_2$, we thus discover that there is just one nontrivial outer automorphism. We first calclulate a matrix representation of 11th irrep by:

```
psi:=Irr(tbl);;
irrepmap:=IrreducibleRepresentationsDixon(g,psi[11]);
```
$$\tag{12}$$

In GAP, an irrep is given by a map from abstract group elements to matrices. To calculate the character of the 11th irrep, we use:

```
List(ConjugacyClasses(g),c->TraceMat(Image(irrepmap,Representative(c))));
```
$$\tag{13}$$

To calculate the character of the 11th irrep after applying the non-trivial automorphism, we use

```
List(ConjugacyClasses(g),c->TraceMat(
    Image(irrepmap,Image(outerautomorphisms[2],Representative(c)))));
```
$$\tag{14}$$

The result turns out to be equal to the characters of the 12th irrep. Therefore, the two 5-dimensional faithful irreps are related by an outer automorphism. We conclusion of this discussion that there exists a single matrix embedding of $S_5 \times \mathbb{Z}_2$ into O(5).

---

[6]If $N$ were an even number, one also need keep complex irreps with dimension $N/2$.

[7]Here `RightCosets(G,H)` calculates the right cosets of $H$ in $G$. The command `Representative(...)` gives the representative elements of the coset, which in our case will be an outer automorphism (excluding the trivial coset). The function `List(ojbects, map)` lists the objects after applying the specified map.

Similarly, we can show that for each of the remaining groups in the list (7) there is a single embedding to consider. As a remark, these groups are easiest specified in GAP as an abstract group by using the "small group ID" (see App. B).

To summarize, we have identified a list of matrix subgroups of O(5) which for sure contains all maximal irreducible subgroups (although not all subgroups in the list are actually maximal irreducible, as we will see). Having such a list is enough to run the group-subgroup relation algorithm to be described in Section 5.

### 4.2 Irreducible Lie subgroups of O(5)

Since we are interested in symmetries of Landau theories, we consider only closed subgroups of O($N$), so any subgroup of interest to us is either finite or a Lie subgroup. We need a separate discussion about irreducible subgroups of O(5) which are Lie subgroups.

The result of the discussion below can be simply stated as follows. There are only three irreducible Lie subgroups of O(5), namely SO(5), SO(3)$_T$ and SO(3)$_T \times \mathbb{Z}_2^{O(5)}$, where $\mathbb{Z}_2^{O(5)}$ is defined in (6), and SO(3)$_T$ is a subgroup obtained by embedding SO(3) into SO(5) using the symmetric traceless 5-dimensional irrep of SO(3), see App. A.

This is argued as follows. First of all since O(5) = SO(5) $\times \mathbb{Z}_2^{O(5)}$, any irreducible Lie subgroup of O(5) must have the form $G$ or $G \times \mathbb{Z}_2^{O(5)}$ where $G$ is an irreducible Lie subgroup of SO(5).[8] We are thus reduced to showing that the only irreducible Lie subgroup of SO(5) is SO(3)$_T$.

Let $H$ be an irreducible Lie subgroup of SO(5), and let $G$ be a maximal Lie subgroup of SO(5) containing $H$. Of course $G$ is also irreducible.

$G$ may be non-simple or simple - we consider these two cases in turn.

1) $G$ is non-simple. Ref. [27] classified all maximal non-simple Lie subgroup of SO($n$) with $n \geq 5$. According to [27], SO(5) has only two such subgroups:

$$S(\mathrm{O}(4) \times \mathrm{O}(1)) \quad \text{and} \quad S(\mathrm{O}(3) \times \mathrm{O}(2)), \tag{15}$$

where

$$S(\mathrm{O}(m) \times \mathrm{O}(n)) := \{(A, B) \in \mathrm{O}(m) \times \mathrm{O}(n), \quad \text{with} \quad \det(A)\det(B) = 1\}, \tag{16}$$

which can be naturally embedded in O($m + n$). Both subgroups (15) are not irreducible (the 5-dimensional representation is reducible). Thus this case is not of interest to us.

2) $G$ is simple. Its Lie algebra should be a simple subalgebra of so(5), which can only be (a compact real form of) so(4), so(3) and so(2). Of these, as a Lie algebra, only so(3) has a 5-dimensional faithful irrep. Thus $G$ must have so(3) as its Lie algebra, and have a 5-dimensional faithful irrep. This singles out SO(3), embedded into SO(5) as the subgroup SO(3)$_T$, see App. A.

So far we proved that $H$ must be SO(3)$_T$ or a subgroup of SO(3)$_T$. All subgroups of SO(3) being known [25, 27], the only Lie subgroups of SO(3) are SO(2) and O(2), and neither of these has a faithful 5-dimensional representation, so this case is excluded. This finishes the proof.

## 5 Group-subgroup relations algorithm

In this section, we discuss an algorithm which given a seed list **groups_initial** of finite irreducible subgroups of O(N), outputs *all* the irreducible finite subgroups of O(N), together with

---

[8]Note that an analogous statement is not true for finite subgroups of O(5). Namely there exist finite subgroups of O(5) which are irreducible yet they are not of the form $G$ or $G \times \mathbb{Z}_2^{O(5)}$ where $G$ is a subgroup of SO(5). An example in $S_6$−$A$ in Fig. 4.

their group-subgroup relations. The seed list must satisfy the following property:

$$\text{\textbf{groups\_initial} contains all finite maximal irreducible subgroups.} \qquad (17)$$

In our case $N = 5$ such a list was obtained in Section 4.

The algorithm is as follows:

---

*Input:*
**groups_initial**=$\{G_1, G_2,\dots G_k\}$ - a list of finite irreducible subgroups of O(N) known to contain all maximal such subgroups.

*Output:*
**groups** - the list of all finite irreducible subgroups of O(N).
**relations** - group-subgroup relations between the groups.

*Algorithm:*
**groups**:=**groups_initial**;
**tocheck**:=**groups_initial**;
**checked**:={};
**relations**:={};
While[**tocheck**$\neq${},
  $G$:=**tocheck**[[1]];
  **subgroups** := all maximal subgroups of $G$;
  **subgroups** := Select[**subgroups**, irreducible subgroups];
  For[$n := 1$, $n \leq$ Length[**subgroups**], $n$++,
   $H$:=**subgroups**[[$n$]];
   **relations**:=**relations** $\cup \{G \supset H\}$;
   If[ $H \notin$ **groups**, **groups**:=**groups** $\cup \{H\}$; ];
   If[ $H \notin$ **checked**, **tocheck**:=**tocheck** $\cup \{H\}$; ];
   **tocheck**:=DeleteElements[**tocheck**, $G$];
  ];
];
Print[**groups**,**relations**];

---

The list **tocheck** stores the groups whose maximal subgroups we still need to find, while the list **checked** stores the groups whose maximal subgroups have been calculated. The code terminates when **tocheck** is empty.

The two lines,

   **subgroups** := all maximal subgroups of $G$;
   **subgroups** := Select[**subgroups**, irreducible subgroups];

are meant to find all the finite irreducible subgroups of group $G$ (up to conjugation). In GAP, this is done as follows. After specifying a matrix group $G$ in GAP (see Eq. (8)), the following command creates a list of maximal matrix subgroups of $G$ (up to conjugation),

$$\texttt{l:=ConjugacyClassesMaximalSubgroups(G);} \qquad (18)$$

The above command generates a list of conjugacy classes of subgroups. Each of these classes contains a set of maximal subgroups which are conjugate to each other. We then check whether these subgroups are irreducible. To do this we compute the character $\chi$ of the $N$-dimensional

representation, and its inner product with itself. The following command does this for the first maximal subgroup in class number `i`:

```
conjugacyclassofsubgroups=l[i];
grp:=conjugacyclassofsubgroups[1];
chi:=Character(grp,List(ConjugacyClasses(grp),
          c->TraceMat(Representative(c))));
ScalarProduct(chi,chi);                                (19)
```

The command `ScalarProduct(chi,chi)` calculates the number $n$ of irreducible representations contained in this 5-dimensional representation by calculating

$$n = \frac{1}{|G|} \sum_{C \in \text{conjugacy classes}} N_C \overline{\chi(C)} \chi(C). \qquad (20)$$

Here $N_C$ is the number of elements in the conjugacy class $C$, and $|G|$ is the order of the group. We get $n = 1$ only if the $N$-dimensional representation remains irreducible **over complex numbers**. When $N$ is an odd number, this is equivalent to being irreducible over reals, which is what we need. When $N$ is an even number, on the other hand, it may happen that an $N$-dimensional representation which are reducible over complex numbers is irreducible over the reals. In that case, one should instead calculate the Molien function and then check that the coefficient of the $z^2$ term is 1. We will discuss the Molien function in Section 7.1. In this paper we have $N = 5$ so we can use the simpler check based on the characters.

Next, we check that $H \notin$ **groups** in the pseudo code. When we do this, we need to check that $H$ is not conjugate in $\mathrm{O}(N)$ to any members of the list **groups**.

Let us explain how such a comparison can be done for any two matrix groups $G_1$ and $G_2$. First we check whether the two matrix groups are isomorphic to each other as abstract groups. This is done as follows. After specifying the groups $G_1$ and $G_2$ using their generators (in terms of $N \times N$ matrices), we try to build an isomorphism between them as abstract groups:

$$\text{iso:=IsomorphismGroups(G1,G2);} \qquad (21)$$

If an isomorphism exist, the output is an isomorphism map, if not, the output just gives "fail". Let us assume an isomorphism iso : $\mathfrak{G}_1 \to \mathfrak{G}_2$ exists. We also have two representations $\rho_1 : \mathfrak{G}_1 \to \mathrm{O}(N)$ and $\rho_2 : \mathfrak{G}_2 \to O(N)$, which are nothing but the matrix groups $G_1$ and $G_2$. Composing the isomorphism and $\rho_2$ we have a second representation $\mathfrak{G}_1$ given by $\rho_2 \circ \text{iso}$. By Lemma 3.1, the two matrix groups $G_1$ and $G_2$ will be conjugate if and only if the representations $\rho_1$ and $\rho_2 \circ \text{iso}$ of $\mathfrak{G}_1$ are equivalent up to outer automorphisms. As we already explained, this can be done by comparing their characters up to the action of the outer automorphism group of $\mathfrak{G}_1$, which can all be computed in GAP.

For example, the characters of $\rho_1$ and $\rho_2 \circ \text{iso}$ are computed by

```
List(ConjugacyClasses(G1),c->TraceMat(Representative(c)));
List(ConjugacyClasses(G1),c->TraceMat(Image(iso,Representative(c))));
```
                                                       (22)

# 6 Results I: Irreducible subgroups of O(5)

Let us explain how we run the above algorithm for O(5). In Section 4, we generated the list of five matrix subgroups of $O(5)$ which contains all maximal irreducible finite subgroups of

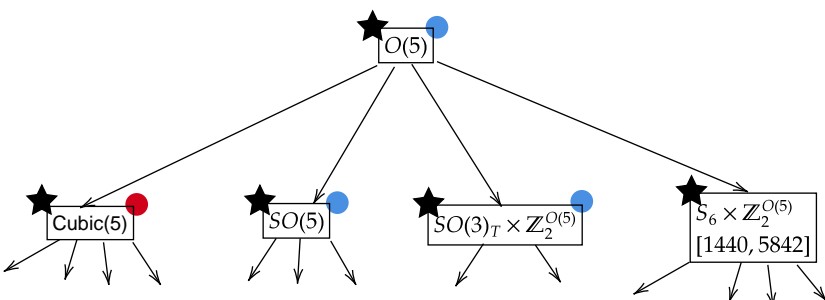

Figure 2: Maximal irreducible subgroups of O(5).

O(5). As abstract groups, they are isomorphic to groups listed in (7), and we discussed how to obtain their matrix embeddings. In addition, as discussed in Section 4.2, O(5) has exactly three irreducible Lie subgroups, namely SO(5), $SO(3)_T$ and $SO(3)_T \times \mathbb{Z}_2^{O(5)}$.

We first run the algorithm from Section 5 with **groups_initial** set to five matrix subgroups corresponding to the list (7). This generates all finite irreducible subgroups of O(5), with their group-subgroup relations. The result is shown in Figs. 2, 3 and 4.

Let us comment on these results starting from Fig. 2. In this figure we see that O(5) has four maximal irreducible subgroups, namely Cubic(5), SO(5), $S_6 \times \mathbb{Z}_2$ and $SO(3)_T \times \mathbb{Z}_2$. To avoid clutter, subgroups of these groups are shown in Figs. 3 and 4. (We will see below how SO(5), $SO(3)_T$ and $SO(3)_T \times \mathbb{Z}_2^{O(5)}$ were positioned in Figs. 3 and 4.)

Let us look at Fig. 3. We see that our algorithm discovered a plethora of finite irreducible subgroups of Cubic(5) which were not in the original list (7), together with their group-subgroup relations.

Groups in Fig. 3 are named according to the structure description in GAP of the corresponding abstract group, which can be easily checked using `StructureDescription(G)`. In Fig. 3 (and also in Appendix B), to be in accord with GAP, we use the notation "$G = N : H$" to denote the (inner) semi-direct product between group $N$ and $H$. In particular, $N$ is a normal subgroup of $G$. This is usually denoted as $N \rtimes H$. Note that the semi-direct product notation does not always fully specify the group; we need to know a homomorphism $\varphi : H \to \mathrm{Aut}(N)$ which may be not unique. The "small group ID" is of course unique.

Another interesting thing to note is that there are three pairs of matrix groups in Fig. 3 which are isomorphic as abstract groups but not conjugate within O(5). E.g. $(\mathbb{Z}_2)^4 : S_5$—A and $(\mathbb{Z}_2)^4 : S_5$—B form such a pair.

Black stars and red/blue dots in Figs. 2, 3 and 4 refer to the Landau condition and RG stability to be discussed in Section 7.

Let us have a similar discussion for Fig. 4. Here, all finite groups are simply related to the groups in the list (7), excluding Cubic(5). This is not too surprising, because the classification of finite subgroups of O(5) used in Section 4.1 treats these subgroups specially.

Let us finally discuss how we inserted SO(5), $SO(3)_T \times \mathbb{Z}_2$ and $SO(3)_T$ in these figures. To position SO(5), we simply check the determinant of the matrix group generators. To position $SO(3)_T \times \mathbb{Z}_2$ and $SO(3)_T$, we use the fact that while $SO(3)_T$ does not have irreducible *Lie* subgroups, it has exactly one irreducible *finite* subgroup, namely $A_5$ (as can be checked using the list of all subgroups of SO(3) [25, 27] plus some other checks). An explicit embedding of $A_5 \subset SO(3)_T$ can be found in Appendix A. This implies unambiguously the group-subgroup relations between $SO(3)_T \times \mathbb{Z}_2$, $SO(3)_T$ and other groups shown in Fig. 4.

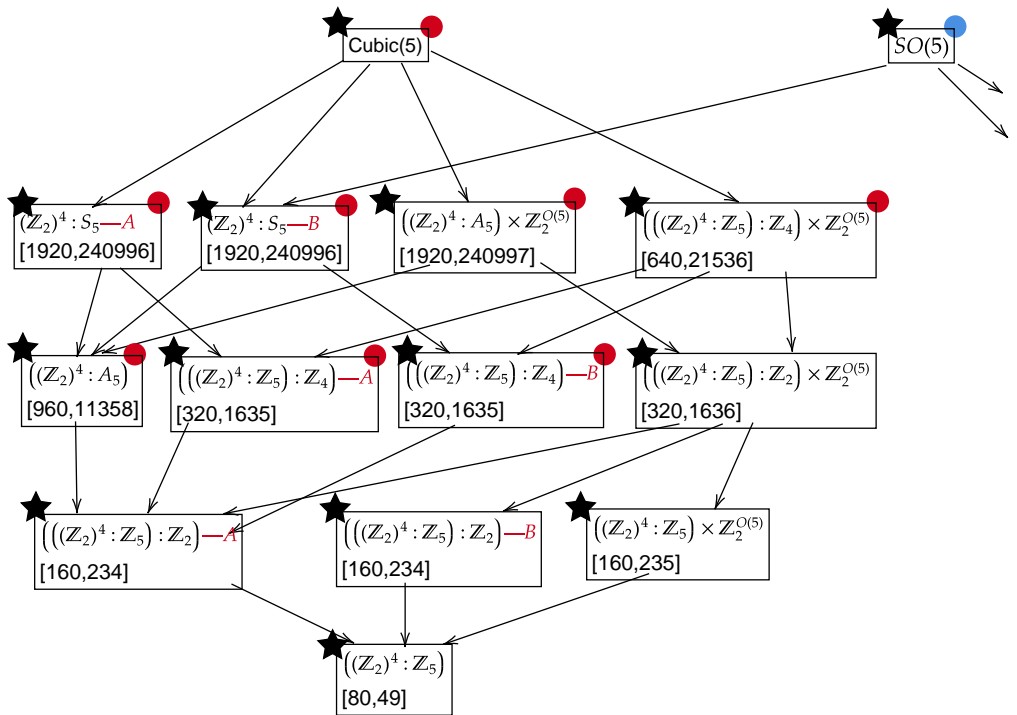

Figure 3: Irreducible subgroups of O(5). The arrows indicates group-subgroup relations: if there is an arrow from $G$ to $H$, this means $H \subset G$ in the sense of matrix groups. Only maximal subgroup relations are indicated by arrows, but note that group-subgroup relations are transitive. The GAP semidirect product notation $N : H$ is equivalent to $N \rtimes H$ more common in physics literature. The numbers in the "[ ]" are "small group ID" of the corresponding abstract group. We use black star to denote that the matrix group satisfies the Landau condition. The groups denoted with red/blue dots are the groups whose Landau theory has a stable fixed point, the color indicating the two different fixed points - see Section 7. The generators and Molien functions of all these groups are given in Appendix B. Note that $\mathbb{Z}_2$ factors for these subgroups do not necessarily act as $\mathbb{Z}_2^{O(5)}$.

# 7  Results II: Landau actions

It remains to analyze the Landau theories corresponding to the irreducible subgroups listed in Figs. 3, 4. As we discussed, we can focus on the Landau theories of the minimal I&L subgroups. Fixed points with larger symmetries, if they exist, will be found as a part of the RG analysis. In this section we work out the Landau theories. RG analysis is performed in the next section.

## 7.1  Landau condition

First we determine which irreducible subgroups satisfy the Landau condition (the absence of cubic invariants). All subgroups containing $\mathbb{Z}_2^{O(5)}$ clearly do not allow cubic invariants. For other groups, this check is done by examining the Taylor expansion of the Molien function, defined as

$$M_\rho(z) = \frac{1}{|G|} \sum_{g \in G} \frac{1}{\det(1 - z\rho(g))}. \tag{23}$$

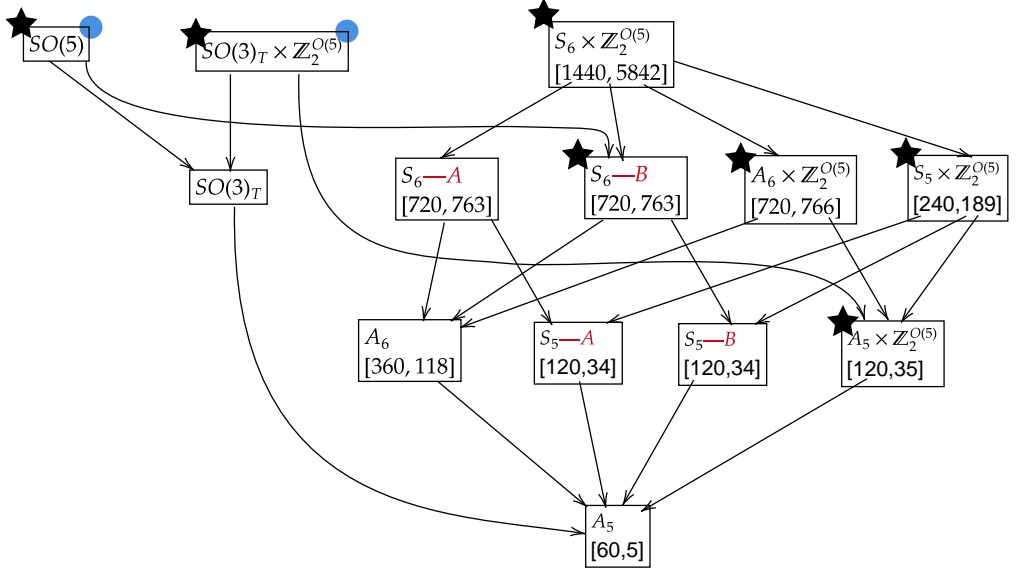

Figure 4: Irreducible subgroups of O(5) (continued), and their group-subgroup relations. Same notation as in Fig. 3.

Here $\rho(g)$ denotes the matrix representation of the group element $g$. In the Taylor expansion of the Molien function, the coefficients of the monomial $z^n$ counts the number of degree-$n$ invariant polynomials. The Landau condition is satisfied if the coefficient of $z^3$ vanishes. The Molien functions of finite groups can be easily calculated using the GAP command

$$\texttt{MolienSeries(chi);} \tag{24}$$

The command takes as an input the character and calculates the Molien function of the corresponding irrep.

The above covers all cases except for $SO(3)_T$. For Lie groups, the role of Molien function is played by the Hilbert series, see for example [28,29]. In our case we don't need this technology, since $SO(3)_T$ clearly has a cubic invariant, given by $t_{ij}t_{jk}t_{ki}$ where $t_{ij}$ is the symmetric traceless tensor of O(3) used to construct the embedding $SO(3)_T \subset O(5)$ in App. A.

We mark the subgroups satisfying the Landau condition by a black star in Figs. 3,4.

We can now identify three minimal I&L subgroups. These are $(\mathbb{Z}_2)^4 : \mathbb{Z}_5$ in Fig. 3 and $S_6{-}B$ and $A_5 \times \mathbb{Z}_2^{O(5)}$ in Fig. 4.

## 7.2 Quartic Landau theories

We will now construct Landau theory for the three minimal I&L subgroups. Let us discuss how to do this for a given matrix subgroup $G \subset O(N)$ explicitly. The quadratic, cubic, and quartic invariant polynomials can be constructed from symmetric tensors $T^{(2)}, T^{(3)}, T^{(4)}$, which are invariant, i.e. satisfy the following equations:

$$
\begin{aligned}
g_{i_1}^{j_1} g_{i_2}^{j_2} T_{j_1 j_2}^{(2)} &= T_{i_1 i_2}^{(2)}, \\
g_{i_1}^{j_1} g_{i_2}^{j_2} g_{i_3}^{j_3} T_{j_1 j_2 j_3}^{(3)} &= T_{j_1 j_2 j_3}^{(3)}, \\
g_{i_1}^{j_1} g_{i_2}^{j_2} g_{i_3}^{j_3} g_{i_4}^{j_4} T_{j_1 j_2 j_3 j_4}^{(4)} &= T_{j_1 j_2 j_3 j_4}^{(4)},
\end{aligned}
\tag{25}
$$

which should be satisfied for all $g \in G$. Since we care about fixed point in $4-\epsilon$ dimension, we truncate the Landau theories to the quartic order and do not consider higher-order invariants.

If $G$ is finite, it's enough to check the above equations for a finite list of generators $g_1, \ldots g_k$. We get a finite list of linear equations, solving which we get a basis of invariants.

Since our subgroup is irreducible, the $T^{(2)}$ equation allows a single solution $T^{(2)}_{ij} = \delta_{ij}$. Because of the Landau condition, the second equation has no nontrivial solutions. Solving the third equation, we construct the quartic Landau theory. The number of solutions to this equation, $I_4$, can be determined from the $z^4$ coefficient of the Molien function, but to find the actual invariant tensors we have to solve this equation.

So we have to do this for $(\mathbb{Z}_2)^4 : \mathbb{Z}_5$ in Fig. 3, and for $S_6{-}B$ and $A_5 \times \mathbb{Z}_2^{O(5)}$ in Fig. 4. The RG analysis will be discussed in the next section.

### 7.2.1 $(\mathbb{Z}_2)^4 : \mathbb{Z}_5$

For $(\mathbb{Z}_2)^4 : \mathbb{Z}_5$, from the Molien series, we know that $I_4 = 3$ - there are three independent invariant quartic polynomials. Using the generators listed in Appendix B, we can calculate them:

$$
\begin{aligned}
P_1(x) &= (x_1^2 + x_2^2 + x_3^2 + x_4^2 + x_5^2)^2, \\
P_2(x) &= x_1^4 + x_2^4 + x_3^4 + x_4^4 + x_5^4, \\
P_3(x) &= x_1^2 x_2^2 + x_2^2 x_3^2 + x_3^2 x_4^2 + x_4^2 x_5^2 + x_5^2 x_1^2.
\end{aligned}
\tag{26}
$$

We parametrize the Landau potential as

$$
V(\phi) = \frac{1}{4!}\lambda_1 P_1(\phi) + \frac{1}{4!}\lambda_2 P_2(\phi) + \frac{1}{4!}\lambda_3 P_3(\phi).
\tag{27}
$$

At this point we can ask the following question. All polynomials in (26) are, by construction, invariant under $(\mathbb{Z}_2)^4 : \mathbb{Z}_5$. But can it be that they are actually invariant under some larger subgroup $G$ containing $(\mathbb{Z}_2)^4 : \mathbb{Z}_5$? As a matter of fact, this indeed happens in our situation. To see that this happens, it is enough to compute the number of quartic invariants $I_4$ for subgroups $G$ in Fig. 4 containing $(\mathbb{Z}_2)^4 : \mathbb{Z}_5$. Doing so one discovers that the three subgroups "one level up", i.e. $((\mathbb{Z}_2)^4 : \mathbb{Z}_5) : \mathbb{Z}_2$, $(((\mathbb{Z}_2)^4 : \mathbb{Z}_5) : \mathbb{Z}_2){-}A$ and $(((\mathbb{Z}_2)^4 : \mathbb{Z}_5) : \mathbb{Z}_2){-}B$ also have $I_4 = 3$. Going one level further up, the group $(((\mathbb{Z}_2)^4 : \mathbb{Z}_5) : \mathbb{Z}_2) \times \mathbb{Z}_2^{O(5)}$ still has $I_4 = 3$, while other subgroups at this level have $I_4 = 2$. On the next level up, all subgroups have $I_4 < 3$.

This computation implies that all polynomials in (26) are also invariant polynomials of the largest of the above-mentioned $I_4 = 3$ subgroups, namely $(((\mathbb{Z}_2)^4 : \mathbb{Z}_5) : \mathbb{Z}_2) \times \mathbb{Z}_2^{O(5)}$. Following Ref. [1], we call **centralizer** the maximal symmetry group of a quartic Landau theory. Any fixed point, if it exists, will have symmetry at least as large as the centralizer, which in the case at hand is strictly larger than the minimal irreducible subgroup we started with. We call this phenomenon "centralizer enhancement".

It's good to have some idea how the discussed groups actually act. The group $(((\mathbb{Z}_2)^4 : \mathbb{Z}_5) : \mathbb{Z}_2) \times \mathbb{Z}_2^{O(5)}$ can be generated by three generators, two of them being permutations of coordinate axes

$$
(1, 2, 3, 4, 5) \quad \text{and} \quad (1, 5)(2, 4).
\tag{28}
$$

Here we are following the cycle notation to denote permutation of the five coordinates. The third generator is given by

$$
X: \quad x_1 \to -x_1, \quad \text{and} \quad x_i \to x_i, \quad \text{for} \quad i \neq 1.
\tag{29}
$$

On the other hand the group $(\mathbb{Z}_2)^4 : \mathbb{Z}_5$ is generated by the axis permutation $(1,2,3,4,5)$ and the following second generator

$$
Y: \quad x_1 \to -x_1, \quad x_2 \to -x_2, \quad x_3 \to -x_3, \quad x_4 \to -x_4, \quad \text{and} \quad x_5 \to x_5
\tag{30}
$$

(one checks easily that $Y \in (((\mathbb{Z}_2)^4 : \mathbb{Z}_5) : \mathbb{Z}_2) \times \mathbb{Z}_2^{O(5)})$. Now, it is easy to see that polynomials (26) are invariant under $(\mathbb{Z}_2)^4 : \mathbb{Z}_5$ and that they are also invariant under the bigger group $(((\mathbb{Z}_2)^4 : \mathbb{Z}_5) : \mathbb{Z}_2) \times \mathbb{Z}_2^{O(5)}$. This provides a check of the above reasoning based on the Molien series.

The phenomenon of centralizer enhancement can be defined generally. If $H \subset G$, and $H$ and $G$ have the same truncated Molien series up to $z^4$, their quartic Landau theories will be equivalent. In other words, there is a basis such that the quartic Landau theories for $H$ and $G$ look exactly the same. Groups related by group-subgroup relations with the same truncated Molien series form a directed graph; they all have equivalent quartic Landau theories. In fact, the symmetry of the quartic Landau action will be given by the maximal vertex of this directed graph.

It turns out that the quartic Landau theories corresponding to all the groups in Fig. 3 all reduce to the following Landau theories (we have included the O(5) group for demonstration purposes)

$$
\begin{aligned}
\text{O(5)} &: \frac{1}{4!}\lambda_1 P_1(x), \\
\text{Cubic(5)} &: \frac{1}{4!}\lambda_1 P_1(x) + \frac{1}{4!}\lambda_2 P_2(x), \\
(((\mathbb{Z}_2)^4 : \mathbb{Z}_5) : \mathbb{Z}_2) \times \mathbb{Z}_2^{O(5)} &: \frac{1}{4!}\lambda_1 P_1(x) + \frac{1}{4!}\lambda_2 P_2(x) + \frac{1}{4!}\lambda_3 P_3(x).
\end{aligned}
\tag{31}
$$

In this table, we label the quartic Landau theories by their maximal symmetry groups i.e. the centralizer. In fact, when we wrote down the invariant polynomials (26), we have taken into account the above structure. In other words, the polynomial $P_1(x)$ preserves the O(5) group, the polynomial $P_2(x)$ breaks the $O(5)$ group to Cubic(5), the polynomial $P_3(x)$ further breaks the Cubic(5) group to $(((\mathbb{Z}_2)^4 : \mathbb{Z}_5) : \mathbb{Z}_2) \times \mathbb{Z}_2^{O(5)}$.

### 7.2.2 $S_6{-}B$ and $A_5 \times \mathbb{Z}_2^{O(5)}$

Due to the centralizer enhancement (see Section 7.2.1), the quartic Landau theory for $S_6{-}B$, $A_5 \times \mathbb{Z}_2^{O(5)}$, and their parents $S_6 \times \mathbb{Z}_2^{O(5)}$ and $A_6 \times \mathbb{Z}_2^{O(5)}$ are all equivalent. This can be seen by computing $I_4$ for these groups. It turns out that $I_4 = 2$ for all these groups, which implies the above statement. So we discuss the Landau theory for $S_6 \times \mathbb{Z}_2^{O(5)}$.

This Landau theory belongs to a family of Landau theories with the tetrahedral symmetry $S_{N+1} \times \mathbb{Z}_2^{O(N)}$ mentioned in Section 2. The Landau theory is constructed by taking $N+1$ vectors $e_i^\alpha \in \mathbb{R}^N$, $\alpha = 1 \ldots N+1$ such that [16]

$$
e_i^\alpha e_i^\beta = \frac{N+1}{N}\delta^{\alpha\beta} - \frac{1}{N}.
\tag{32}
$$

The endpoints of these $N + 1$ vectors are vertices of a hypertetrahedron in an $N$-dimensional Euclidean space. In addition to the O($N$) invariant quartic coupling, the only other quartic coupling allowed by the symmetry is $\left(\sum_\alpha e_i^\alpha e_j^\alpha e_k^\alpha e_l^\alpha\right)\phi^i\phi^j\phi^k\phi^l$. The cubic term $\left(\sum_\alpha e_i^\alpha e_j^\alpha e_k^\alpha\right)\phi_i\phi_j\phi_k$ is forbidden by $\mathbb{Z}_2^{O(N)}$.

Applying this construction for $N = 5$, we get the $S_6 \times \mathbb{Z}_2^{O(5)}$ Landau theory

$$
S_6 \times \mathbb{Z}_2^{O(5)} : \frac{1}{4!}\lambda_1 P_1(\phi) + \frac{1}{4!}\lambda_4 P_4(\phi),
\tag{33}
$$

with $P_1(x)$ defined in (26) and $P_4(\phi) = \left(\sum_\alpha e_i^\alpha e_j^\alpha e_k^\alpha e_l^\alpha\right)\phi^i\phi^j\phi^k\phi^l$.[9]

---

[9]This polynomial is not particularly beautiful so we don't report it.

To summarize, the Landau theories in Eq. (31) and Eq. (33) exhaust all the quartic Landau theories that can potentially give us an irreducible fixed point for $N = 5$ in $d = 4-\epsilon$ dimensions. We will now proceed to discuss their RG flows.

# 8   Results III: RG analysis

At the beginning of this paper we stated our goal as classifying all real fixed points with $N = 5$ having symmetry $G$ and satisfying the conditions I (irreducibility), L (Landau) and S (stability).

Our RG analysis will allow us to solve this goal, but at the same time to provide a more nuanced information concerning symmetry enhancement, which we will now describe.

## 8.1   Stability of fixed points

In the introduction we described stability S as irrelevance of all quartic perturbations preserving the fixed point symmetry $G$. More generally, we will say that a fixed point is $H$-stable, if all quartic perturbations preserving symmetry $H$ are irrelevant. Otherwise we call it $H$-unstable. Here $H$ may be a subgroup of the fixed point symmetry $G$.

This general definition is interesting, because it may happen that an RG flow having smaller symmetry $H$ leads to a stable fixed point having larger symmetry $G$. This is called symmetry enhancement. For this to happen without additional fine-tuning, the fixed point has to be $H$-stable.

Having this application in mind, we will only consider $H$ which are themselves irreducible and satisfy the Landau condition.

In general, to discuss $H$-stability of a fixed point, one has to realize it within the Landau theory having symmetry $H$, and compute the eigenvalues of the stability matrix

$$\frac{\partial \beta_i}{\partial \lambda_j},\tag{34}$$

at the fixed point, where $\beta_i$ are the beta functions and $\lambda_i$ are the independent couplings. The fixed point is $H$-stable if all the eigenvalues are positive.

As we have seen in Section 7.2.1, often two groups $H_1 \subset H_2$ have the same Landau theories (centralizer enhancement). If so, stability properties of a fixed point with respect to $H_2$ and $H_1$ will be exactly the same. For us this means that we need to analyze $H$-stability only with respect to 4 groups which are the symmetry groups of the four Landau theories listed in Eqs. (31) and Eq. (33).

Given that some fixed points will occur repeatedly below, let us discuss their stability here. The first fixed point is the free theory, which has $G = O(N)$ invariance. This fixed point is O($N$)-unstable (and hence $H$-unstable for any $H \subset O(N)$). This follows from considering the beta function for the single O($N$)-invariant coupling.

Then, there is the O($N$) Wilson-Fisher fixed point, which also has $G = O(N)$. This fixed point is O($N$)-stable, the eigenvalue given by

$$\omega = \varepsilon + O(\varepsilon^2).\tag{35}$$

For $N \geqslant 5$, it is unstable with respect to any smaller $H$. This was first shown in [5]. The proof is as follows. Let us decompose all quartic perturbations of the O($N$) Wilson-Fisher fixed point with respect to the fixed point symmetry O($N$). In addition to the O($N$) invariant perturbation, there are perturbations in symmetric traceless representations with 2 and 4 indices. 2-index perturbations are not allowed by symmetry $H$ if it is irreducible, as we assume. Some 4-index perturbations may be allowed by $H$, but whatever $H$ is, all of them will have exactly the same

stability matrix eigenvalue at the fixed point. This eigenvalue can be computed using any symmetric traceless 4-index perturbation, and it is given by [5] (see also [30], Table 25, line 1),

$$\omega' = \frac{N-4}{N+8}\varepsilon + O(\varepsilon^2).\tag{36}$$

## 8.2 $(((\mathbb{Z}_2)^4 : \mathbb{Z}_5) : \mathbb{Z}_2) \times \mathbb{Z}_2^{O(5)}$ invariant quartic Landau theory

We now proceed to the RG analysis of the $(((\mathbb{Z}_2)^4 : \mathbb{Z}_5) : \mathbb{Z}_2) \times \mathbb{Z}_2^{O(5)}$ invariant quartic Landau theory given in Eq. (31). Recall that this theory was originally derived for the symmetry group $(\mathbb{Z}_2)^4 : \mathbb{Z}_5$, but because of the centralizer enhancement, the symmetry of the quartic Landau theory turned out to be larger.

Using the general one-loop beta functions (2), we get beta functions for the couplings $\lambda_1, \lambda_2, \lambda_3$:

$$\begin{aligned}
\beta_1 &= -\epsilon\lambda_1 + \frac{13\lambda_1^2}{48\pi^2} + \frac{\lambda_2\lambda_1}{8\pi^2} + \frac{\lambda_3\lambda_1}{24\pi^2} + \frac{\lambda_3^2}{192\pi^2}\,,\\
\beta_2 &= -\epsilon\lambda_2 + \frac{3\lambda_2^2}{16\pi^2} + \frac{\lambda_1\lambda_2}{4\pi^2} + \frac{\lambda_3^2}{192\pi^2}\,,\\
\beta_3 &= -\epsilon\lambda_3 + \frac{\lambda_3^2}{32\pi^2} + \frac{\lambda_2\lambda_3}{8\pi^2} + \frac{\lambda_1\lambda_3}{4\pi^2}\,.
\end{aligned}\tag{37}$$

They have in total 4 fixed points with real couplings:

$$\begin{aligned}
F_1: \quad & \lambda_1 = \lambda_2 = \lambda_3 = 0\,,\\
F_2: \quad & \lambda_1 = \frac{48\pi^2}{13}\epsilon, \lambda_2 = 0, \lambda_3 = 0\,,\\
F_3: \quad & \lambda_1 = 0, \lambda_2 = \frac{16\pi^2}{3}\epsilon, \lambda_3 = 0\,,\\
F_4: \quad & \lambda_1 = \frac{16\pi^2}{5}\epsilon, \lambda_3 = \frac{16\pi^2}{15}\epsilon, \lambda_3 = 0\,.
\end{aligned}\tag{38}$$

Let us discuss the physical interpretation. Fixed points $F_1$ and $F_2$ preserve full O(5), these are the free theory and the O(5) Wilson-Fisher fixed point, at the lowest order in $\epsilon$. Fixed points $F_3$ and $F_4$ preserve the Cubic(5) subgroup. Fixed point $F_3$ is interpreted as 5 decoupled copies of the Ising model fixed point. Fixed point $F_4$ is the $N = 5$ case of the Cubic($N$) fixed point first mentioned in Section 2.

In addition to the above 4 real fixed points, we have 4 fixed points with complex couplings:

$$F_5: \quad \lambda_1 = \frac{8}{7}\left(3 - i\sqrt{3}\right)\pi^2\epsilon, \quad \lambda_2 = \frac{8}{7}\left(1 + i\sqrt{3}\right)\pi^2\epsilon, \quad \lambda_3 = \frac{32}{7}i\sqrt{3}\pi^2\epsilon,\tag{39}$$

$$F_7: \quad \lambda_1 = \frac{8}{5}\left(3 - i\sqrt{3}\right)\pi^2\epsilon, \quad \lambda_2 = \frac{8}{5}i\left(\sqrt{3} + i\right)\pi^2\epsilon, \quad \lambda_3 = \frac{32}{5}i\sqrt{3}\pi^2\epsilon,\tag{40}$$

and their complex conjugated partners $F_6 = F_5^*$ and $F_8 = F_7^*$. These fixed points have symmetry $(((\mathbb{Z}_2)^4 : \mathbb{Z}_5) : \mathbb{Z}_2) \times \mathbb{Z}_2^{O(5)}$.[10] Since the above mentioned fixed points are complex they are not of our primary physical interest.

---

[10]Considering a finite set of polynomials $\{P_i\}$, the symmetry group of a specific linear combination of such polynomials $\sum_i \lambda_i P_i$ is defined as the group that leaves it invariant. This is also sometimes referred to as the little group. The symmetry group, as defined, may depend on the coefficients $\lambda_i$. The centralizer group, on the other hand, is the intersection of the symmetry groups of all such polynomials. Usually the symmetry group of a *generic* polynomial coincides with the centralizer group. In rare cases this is violated and a generic polynomial has a strictly larger symmetry group. If this happens, one expects to finds continuous families of fixed points rather than isolated fixed points [1]. In our case there are only isolated fixed points, and so the symmetry group of all fixed points coincides with the centralizer of the corresponding polynomial family.

Let us verify fixed point counting. We are solving 3 quadratic polynomial equations $\beta_i = 0$, with $i = 1, 2, 3$. Generically the number of solutions equals the product of degrees of the polynomials [31], and that's what we have here, which means that every solution has multiplicity one. Mathematically this means that the Jacobian matrix $\frac{\partial \beta_i}{\partial \lambda_j}$ is non-degenerate (the determinant is nonzero) for every of the above solutions, as one can easily check. This matrix is the same as the stability matrix mentioned above. Its eigenvalues for fixed points $F_1, \ldots, F_4$ will be reported below, and we will see that they are all nonzero.

The above analysis was based on the one-loop beta function equations, and one may ask what will happen with the above solutions when we add further loops. In the case at hand, higher loops will not change the picture. The above solutions will be corrected by higher integer powers of $\epsilon$, but the number of solutions will not change and their symmetry will remain the same. This is so precisely because all one-loop solutions have multiplicity one, and the Jacobian is non-degenerate.[11]

We note that we are not interested here in the solutions of higher-order beta functions equations whose couplings do not go to zero as $\epsilon \to 0$.

For every fixed point in the above list, we will now examine if it is $H$-stable with respect to some $H \subset G$ where $G$ is the fixed point symmetry. As discussed in Section 8.1, we may restrict our attention to $H$ being one of the four groups which are symmetries of the Landau theories in Eq. (31) and Eq. (33). Any other $H$ will have Landau theory equivalent to one of these four theories, and the stability properties will be the same. Eq. (33) will be studied in the next subsection; here we are interested in Eq. (31).

At $F_1$ (free theory), $G = $ O(5), the matrix has three degenerate eigenvalues with $\omega_1 = \omega_2 = \omega_3 = -\epsilon$. One of the eigenvectors is O(5) invariant. This means that this fixed point is unstable even if we take the largest $H = G = $ O(5). We have already discussed this instability generally in Section 8.1.

At $F_2$ (O(5) Wilson-Fisher fixed point), we have

$$
\begin{array}{lll}
\omega_1 = \epsilon\,, & P_1(\phi)\,, & \text{O}(5)\,, \\
\omega_2 = -\dfrac{1}{13}\epsilon\,, & P_1(\phi) - 7P_2(\phi)\,, & \text{Cubic}(5)\,, \\
\omega_3 = -\dfrac{1}{13}\epsilon\,, & 3P_1(\phi) - 7P_3(\phi)\,, & (((\mathbb{Z}_2)^4 : \mathbb{Z}_5) : \mathbb{Z}_2) \times \mathbb{Z}_2^{O(5)}\,,
\end{array}
\tag{41}
$$

where the second column shows the eigenvectors and the third columns shows the largest $H$ which allows this perturbation. We see that this fixed point is $H$-stable only if $H = G = $ O(5). This agrees with the general discussion in Section 8.1. The degeneracy of $\omega_2$ and $\omega_3$ has a simple reason, also discussed there — the corresponding eigenvectors belong to the same irrep of O(5), that is the 4-index symmetric traceless tensor irrep.

At $F_3$ (decoupled Ising models), the symmetry group of the fixed point is $G = $ Cubic(5). We have

$$
\begin{array}{lll}
\omega_1 = \epsilon\,, & P_2(\phi)\,, & \text{Cubic}(5)\,, \\
\omega_2 = -\dfrac{1}{3}\epsilon\,, & P_1(\phi) - P_2(\phi)\,, & \text{Cubic}(5)\,, \\
\omega_2 = -\dfrac{1}{3}\epsilon\,, & P_3(\phi)\,, & (((\mathbb{Z}_2)^4 : \mathbb{Z}_5) : \mathbb{Z}_2) \times \mathbb{Z}_2^{O(5)}\,.
\end{array}
\tag{42}
$$

We conclude that this fixed point is unstable even if $H = G = $ Cubic(5).

---

[11]Mathematically, this is easy to see by applying the implicit function theorem after rescaling the couplings $g \to \epsilon g$. For an explicit discussion, see [6].

At $F_4$ (the cubic fixed point) we also have $G = \text{Cubic}(5)$ and

$$
\begin{aligned}
\omega_1 &= \epsilon\,, & 3P_1(\phi) + P_2(\phi)\,, & & \text{Cubic}(5)\,, \\
\omega_2 &= \frac{1}{15}\epsilon\,, & P_1(\phi) - 2P_2(\phi)\,, & & \text{Cubic}(5)\,, \\
\omega_2 &= -\frac{1}{13}\epsilon\,, & P_1(\phi) - P_2(\phi) - 4P_3(\phi)\,, & & (((\mathbb{Z}_2)^4 : \mathbb{Z}_5) : \mathbb{Z}_2) \times \mathbb{Z}_2^{O(5)}\,.
\end{aligned}
\tag{43}
$$

Clearly this fixed point is $H$-stable for $H = G = \text{Cubic}(5)$, and is unstable for smaller $H$. This fixed point belongs to the family of fixed points with $\text{Cubic}(N)$ symmetry discussed in Section 2.

## 8.3 $S_6 \times \mathbb{Z}_2^{O(5)}$ invariant quartic Landau theory

We now move to the Landau theory (33). The one-loop beta functions are

$$
\begin{aligned}
\beta_1(\lambda_1, \lambda_2) &= -\lambda_1 \epsilon + \frac{13\lambda_1^2}{48\pi^2} + \frac{3\lambda_1\lambda_2}{20\pi^2} + \frac{27\lambda_2^2}{2500\pi^2}\,, \\
\beta_2(\lambda_1, \lambda_2) &= -\lambda_2 \epsilon + \frac{\lambda_1\lambda_2}{4\pi^2} + \frac{9\lambda_2^2}{50\pi^2}\,.
\end{aligned}
\tag{44}
$$

Solving the beta function equations we have three real fixed points at one-loop:

$$
\begin{aligned}
F_1: & \quad \lambda_1 = \lambda_2 = 0\,, \\
F_2: & \quad \lambda_1 = \frac{48\pi^2}{13}\epsilon\,, \quad \lambda_2 = 0\,, \\
F_9, F_{10}: & \quad \lambda_1 = 2\pi^2\epsilon\,, \lambda_2 = \frac{25\pi^2}{9}\epsilon\,.
\end{aligned}
\tag{45}
$$

The first two fixed points are the free theory and the O(5) fixed point discussed in Section 8.2. The third fixed point is a new fixed point having $S_6 \times \mathbb{Z}_2^{O(5)}$ symmetry. It's actually a pair of fixed points which are degenerate at this order, since we expect four fixed points by Bezout's theorem. To resolve their degeneracy we go to the two-loop order. The two-loop beta functions for the general quartic coupling $\frac{1}{4!}\lambda_{ijkl}\phi^i\phi^j\phi^k\phi^l$ takes the form [5, 32, 33][12]

$$
\beta_{ijkl}^{2\text{nd}}(\lambda) = \frac{1}{(4\pi)^4}\left(\frac{1}{12}\lambda_{ijkl}(H_{ii} + H_{jj} + H_{kk} + H_{ll}) - \frac{1}{4}\sum_{\text{perms}}\sum_{mnpr}\lambda_{ijmn}\lambda_{kmpr}\lambda_{lnpr}\right)\,,
\tag{46}
$$

where

$$
H_{ij} = \sum_{mnp}\lambda_{imnp}\lambda_{jmnp}\,.
\tag{47}
$$

We do not assume the Einstein summation convention in the above formulas. The summation $\sum_{\text{perms}}$ sums over the 24 permutations of $i, j, k, l$ indices.

In our case we get

$$
\begin{aligned}
\beta_1^{2\text{nd}} &= -\frac{29\lambda_1^3}{768\pi^4} - \frac{11\lambda_2\lambda_1^2}{320\pi^4} - \frac{423\lambda_2^2\lambda_1}{40000\pi^4} - \frac{81\lambda_2^3}{62500\pi^4}\,, \\
\beta_2^{2\text{nd}} &= -\frac{171\lambda_2^3}{8000\pi^4} - \frac{97\lambda_1\lambda_2^2}{1600\pi^4} - \frac{107\lambda_1^2\lambda_2}{2304\pi^4}\,.
\end{aligned}
\tag{48}
$$

---

[12]We note that the beta function for the general quartic couplings has recently been obtained up to six loops [34].

Adding this correction to the one-loop beta function, we find that the pair $F_9, F_{10}$ becomes a pair of complex conjugate fixed points

$$F_9, F_{10}: \quad \lambda_1 = 2\pi^2 \epsilon \mp \sqrt{6}\pi^2 i \epsilon^{3/2} + \mathrm{O}(\epsilon^2), \quad \lambda_2 = \frac{25\pi^2}{9}\epsilon \pm \frac{25\pi^2}{3\sqrt{6}} i \epsilon^{3/2} + \mathrm{O}(\epsilon^2). \quad (49)$$

The same fractional power of $\epsilon$ appears in the eigenvalues of the stability matrix.

This discussion leaves us only two real fixed points, the free theory fixed point and the O(5) Wilson-Fisher fixed point. Their stability properties have already been discussed in the previous sections.

## 9 Discussion

In this paper we classified all fixed points with $N = 5$ scalar fields in $4 - \varepsilon$ dimensions whose symmetry group $G \subset \mathrm{O}(N)$ is irreducible (thus allowing a single mass term) and satisfies the Landau condition (thus forbidding the cubic terms). In addition, we impose the constraints of RG stability and reality. In the end we found only two fixed points satisfying all these conditions: the Wilson-Fisher O(5) and the cubic fixed point having Cubic(5) symmetry. Both of these fixed points are of course well known. So our main result is that no further fixed points of this type exist for $N = 5$.

It's interesting to compare our analysis to the work of Toledano et al [1], who in 1985 solved the analogous problem for $N = 4$. They had to work much harder than us, for several reasons. Their list of irreducible subgroups, group-subgroup relations, and Landau theories turned to be much longer than for us. Many of their fixed points were degenerate at one-loop, necessitating to go to the two-loop order. The results of their beautiful analysis were somewhat disappointing, as for us, since they discovered no new stable fixed points - all 4 stable fixed points of their final list have already been known before their work.

It is important however not to despair and to continue this program to higher values of $N$. The $N = 6$ case appears within reach, because finite subgroups of SU(4) have been classified [35,36]. It would be very exciting if exhaustive analysis yielded a new stable fixed point missed by prior studies.

One may also consider an analogous problem for more general theories, including fermions and gauge fields. Recall that the two-loop beta functions for the most general renormalizable quantum field theories in four dimension (quartic scalar, Yukawa, and gauge couplings) are known since the 1980's [37–39]. There exist many known fixed points of these theories, in particular of the scalar-fermion theories [40–44]. It will be interesting to systematically study fixed points of these theories in $d = 4 - \varepsilon$, with the help of group theory results. Notice that the classification of finite subgroups of many Lie groups are also finished, including SU(3) [35,45] and SU(4) mentioned above.

Another variation of our problem would be to try to classify tri-critical fixed points, which means fixed points with two relevant couplings (which may be two mass terms, one mass and one cubic, or one mass and one quartic).

In this paper we only worked in perturbation theory around $d = 4$. It is interesting to inquire what may happen when the fixed points we discussed are continued to $\epsilon = \mathrm{O}(1)$. Some stability results we mentioned are known to be robust with respect to such continuation. For example, the O(5) Wilson-Fisher fixed point is stable in any $2 < d < 4$. However sometimes two fixed points collide in some intermediate dimension and exchange their stability properties. Two such examples were mentioned in Remark 2.1.

There is also another way how physics near 4 dimensions may be different from the physics in 3 dimensions. We found several fixed points, with interesting symmetries, which had complex couplings at small $\epsilon$. More precisely these were pairs of complex-conjugate fixed points.

(Similar fixed points were also found in [1].) One may wonder if at $\epsilon = O(1)$ some of these pairs can merge and come to the real plane.

Finally, we mention that although in this paper we focused on the ILS fixed points, as the most important ones for physics applications, it's also mathematically interesting to classify *all* fixed points for a given $N$, independently of their symmetry and stability properties. For recent work in this direction see [7, 14, 46–48]. One open problem is whether, for some $N$, there exists a scalar fixed point with smallest possible symmetry, which is $\mathbb{Z}_2$. A Dom Pérignon champagne bottle offered for solving this problem in [7] remains so far unclaimed.

# Acknowledgments

**Funding information** This work is supported by the Simons Foundation grant 733758 (Simons Bootstrap Collaboration).

# A Embedding SO(3) in SO(5)

In this section we discuss an embedding of SO(3) into SO(5) which gives an irreducible subgroup, which we call SO(3)$_T$. SO(3) has a five dimensional representation which is the symmetric traceless two-index tensor representation. The action is

$$t \to t' = R^T . t . R, \tag{A.1}$$

where $R \in$ SO(3) and $t$ is a $3 \times 3$ symmetric traceless matrix. Parametrizing it as

$$t_{ij} = \begin{pmatrix} x_1 & x_3 & x_4 \\ x_3 & x_2 & x_5 \\ x_4 & x_5 & -x_1 - x_2 \end{pmatrix}. \tag{A.2}$$

We get, for each $R$, a 5-dimensional matrix

$$\tilde{R} = \frac{\partial x'_i}{\partial x_j}. \tag{A.3}$$

This is not yet an embedding into SO(5) because the matrix $\tilde{R}$ is not orthogonal.

The bilinear form which is preserved by $\tilde{R}$ is

$$t_{ij} t_{ij} = x^t g x, \quad g = \begin{pmatrix} 4 & 2 & 0 & 0 & 0 \\ 2 & 4 & 0 & 0 & 0 \\ 0 & 0 & 4 & 0 & 0 \\ 0 & 0 & 0 & 4 & 0 \\ 0 & 0 & 0 & 0 & 4 \end{pmatrix}. \tag{A.4}$$

In other words, we have

$$\tilde{R} . g . \tilde{R}^T = g. \tag{A.5}$$

Let us perform the Cholesky decomposition:

$$g = L^t . L, \quad \text{with} \quad L = \begin{pmatrix} 2 & 1 & 0 & 0 & 0 \\ 0 & \sqrt{3} & 0 & 0 & 0 \\ 0 & 0 & 2 & 0 & 0 \\ 0 & 0 & 0 & 2 & 0 \\ 0 & 0 & 0 & 0 & 2 \end{pmatrix}. \tag{A.6}$$

We define the new $5 \times 5$ matrices by a change of basis:

$$R^{O(5)} = (L^{-1})^t . \tilde{R} . L^t . \tag{A.7}$$

The map $R \rightarrow R^{O(5)}$ is still a representation of SO(3). In addition it is orthogonal – we have $(R^{O(5)})^t . R^{O(5)} = \text{Id}$. Thus we obtained the embedding of SO(3) into SO(5), which we call $SO(3)_T$.

It is easy to work out the images of SO(3) generators under the above embedding. We have

$$J_1 = \begin{pmatrix} 0 & 1 & 0 \\ -1 & 0 & 0 \\ 0 & 0 & 0 \end{pmatrix} \rightarrow \begin{pmatrix} 0 & 0 & -1 & 0 & 0 \\ 0 & 0 & \sqrt{3} & 0 & 0 \\ 1 & -\sqrt{3} & 0 & 0 & 0 \\ 0 & 0 & 0 & 0 & -1 \\ 0 & 0 & 0 & 1 & 0 \end{pmatrix}, \tag{A.8}$$

$$J_2 = \begin{pmatrix} 0 & 0 & 0 \\ 0 & 0 & 1 \\ 0 & -1 & 0 \end{pmatrix} \rightarrow \begin{pmatrix} 0 & 0 & 0 & 0 & -1 \\ 0 & 0 & 0 & 0 & -\sqrt{3} \\ 0 & 0 & 0 & -1 & 0 \\ 0 & 0 & 1 & 0 & 0 \\ 1 & \sqrt{3} & 0 & 0 & 0 \end{pmatrix}, \tag{A.9}$$

$$J_3 = \begin{pmatrix} 0 & 0 & 1 \\ 0 & 0 & 0 \\ -1 & 0 & 0 \end{pmatrix} \rightarrow \begin{pmatrix} 0 & 0 & 0 & -2 & 0 \\ 0 & 0 & 0 & 0 & 0 \\ 0 & 0 & 0 & 0 & -1 \\ 2 & 0 & 0 & 0 & 0 \\ 0 & 0 & 1 & 0 & 0 \end{pmatrix}. \tag{A.10}$$

In Section 6, we mentioned that $SO(3)_T$ group has a maximal subgroup which is isomorphic to $A_5$. This subgroup is generated by

$$g_1 = \begin{pmatrix} \frac{1}{16}(-3\sqrt{5}-1) & -\frac{1}{16}\sqrt{3}(\sqrt{5}-1) & \frac{\sqrt{5}}{4} & \frac{1}{8}(\sqrt{5}+3) & \frac{1}{8}(3-\sqrt{5}) \\ -\frac{1}{16}\sqrt{3}(\sqrt{5}-1) & \frac{1}{16}(3\sqrt{5}+1) & \frac{\sqrt{3}}{4} & -\frac{1}{8}\sqrt{3}(\sqrt{5}-1) & -\frac{1}{8}\sqrt{3}(\sqrt{5}+1) \\ \frac{\sqrt{5}}{4} & \frac{\sqrt{3}}{4} & \frac{1}{2} & 0 & \frac{1}{2} \\ \frac{1}{8}(\sqrt{5}+3) & -\frac{1}{8}\sqrt{3}(\sqrt{5}-1) & 0 & \frac{1}{2} & -\frac{1}{2} \\ \frac{1}{8}(3-\sqrt{5}) & -\frac{1}{8}\sqrt{3}(\sqrt{5}+1) & \frac{1}{2} & -\frac{1}{2} & 0 \end{pmatrix},$$

$$g_2 = \begin{pmatrix} \frac{1}{16}(3\sqrt{5}-1) & -\frac{1}{16}\sqrt{3}(\sqrt{5}+1) & \frac{1}{8}(\sqrt{5}+3) & \frac{1}{8}(3-\sqrt{5}) & \frac{\sqrt{5}}{4} \\ -\frac{1}{16}\sqrt{3}(\sqrt{5}+1) & \frac{1}{16}(1-3\sqrt{5}) & -\frac{1}{8}\sqrt{3}(\sqrt{5}-1) & -\frac{1}{8}\sqrt{3}(\sqrt{5}+1) & \frac{\sqrt{3}}{4} \\ \frac{1}{8}(\sqrt{5}+3) & -\frac{1}{8}\sqrt{3}(\sqrt{5}-1) & 0 & -\frac{1}{2} & -\frac{1}{2} \\ \frac{1}{8}(\sqrt{5}-3) & \frac{1}{8}\sqrt{3}(\sqrt{5}+1) & \frac{1}{2} & -\frac{1}{2} & 0 \\ -\frac{\sqrt{5}}{4} & -\frac{\sqrt{3}}{4} & \frac{1}{2} & 0 & -\frac{1}{2} \end{pmatrix}. \tag{A.11}$$

## B Generators and Molien functions

We list here the generators and Molien functions (see (23)) for all the subgroups listed in Fig. 3 and Fig. 4. Some of the generators below are calculated using GAP, asking for generators of an abstract group in a 5-dimensional irrep. GAP gives generators which are not yet orthogonal matrices. This does not affect the character calculation. If we wish to explicitly construct the invariant tensors of the group, as in Section 7, we need to change to the orthogonal basis. To

achieve that, on need to first calculate the quadratic invariant tensor using the first equation in (25), and then perform the Cholesky decomposition

$$T^{(2)} = L^t . L.$$

The new generators $g'_a = L^{-1,t} . g_a . L^t$ are orthogonal matrices.

We also give "small group ID" of the groups, if they have one. The Small Groups library is a library of GAP that allows us to work with groups of small order conveniently. The "small group ID" is a list of two integers: the first integer equals the order of the group, while the second integer serves to distinguish groups with the same order. Given the ID, one can easily specify the abstract group in GAP by a command like

$$\text{G1:=SmallGroup(1920,240996);} \tag{B.1}$$

To find the generators of an abstract group, with their relations, we use

$$\text{gens:=SmallGeneratingSet(G1);} \tag{B.2}$$

(These are not yet matrix group generators. These may be abstract generators with their relations, or a list of cycles if the group was defined as a group of permutations.)

We have explained how to calculate the character table of the group in (9). To calculate a matrix representation, one needs to first extract the group characters from the character table:

$$\text{irreps:=Irr(tbl);} \tag{B.3}$$

The following command then calculates (generators of) the 7th irreducible representations of the abstract group $\mathfrak{G}_1$,

$$\text{hom:=IrreducibleRepresentationsDixon(G1, irreps[7]);} \tag{B.4}$$

The output of the above command is an isomorphism from the abstract group $\mathfrak{G}_1$ to a matrix group. To find the actual matrix representation of the generators we use

$$\text{List(gens, c->Image(hom, c));} \tag{B.5}$$

After this preamble, we give the generators and the Molien series. We start with the finite irreducible subgroups, and consider the Lie subgroups below.

1. Cubic(5):

$$\begin{pmatrix} -1 & 0 & 0 & 0 & 0 \\ 0 & 1 & 0 & 0 & 0 \\ 0 & 0 & 1 & 0 & 0 \\ 0 & 0 & 0 & 1 & 0 \\ 0 & 0 & 0 & 0 & 1 \end{pmatrix}, \begin{pmatrix} 0 & 0 & 0 & 0 & 1 \\ 1 & 0 & 0 & 0 & 0 \\ 0 & 1 & 0 & 0 & 0 \\ 0 & 0 & 1 & 0 & 0 \\ 0 & 0 & 0 & 1 & 0 \end{pmatrix}, \begin{pmatrix} 0 & 1 & 0 & 0 & 0 \\ 1 & 0 & 0 & 0 & 0 \\ 0 & 0 & 1 & 0 & 0 \\ 0 & 0 & 0 & 1 & 0 \\ 0 & 0 & 0 & 0 & 1 \end{pmatrix}, \tag{B.6}$$

$$M_5(z) = \frac{1}{(1-z^2)(1-z^4)(1-z^6)(1-z^8)(1-z^{10})}. \tag{B.7}$$

2. $(\mathbb{Z}_2)^4 : S_5 - A$, [1920,240996]:

$$\begin{pmatrix} 0 & 0 & 0 & 0 & 1 \\ 0 & 0 & 0 & -1 & 0 \\ 0 & 0 & -1 & 0 & 0 \\ -1 & 0 & 0 & 0 & 0 \\ 0 & -1 & 0 & 0 & 0 \end{pmatrix}, \begin{pmatrix} 0 & 0 & 0 & -1 & 0 \\ 0 & -1 & 0 & 0 & 0 \\ -1 & 0 & 0 & 0 & 0 \\ 0 & 0 & 1 & 0 & 0 \\ 0 & 0 & 0 & 0 & -1 \end{pmatrix}, \tag{B.8}$$

$$M_5(z) = \frac{1}{(1-z^2)(1-z^4)(1-z^5)(1-z^6)(1-z^8)}. \tag{B.9}$$

3. $(\mathbb{Z}_2)^4 : S_5 - B$, $[1920, 240996]$:

$$
\begin{pmatrix}
0 & 0 & 0 & 0 & -1 \\
0 & 1 & 0 & 0 & 0 \\
0 & 0 & -1 & 0 & 0 \\
0 & 0 & 0 & -1 & 0 \\
1 & 0 & 0 & 0 & 0
\end{pmatrix},
\begin{pmatrix}
0 & 0 & 0 & 0 & 1 \\
0 & 0 & 1 & 0 & 0 \\
0 & 0 & 0 & -1 & 0 \\
1 & 0 & 0 & 0 & 0 \\
0 & -1 & 0 & 0 & 0
\end{pmatrix},
\tag{B.10}
$$

$$
M_5(z) = \frac{z^{20} - z^{15} + z^{10} - z^5 + 1}{(1-z^2)(1-z^4)(1-z^5)(1-z^6)(1-z^8)} .
\tag{B.11}
$$

4. $((\mathbb{Z}_2)^4 : A_5) \times \mathbb{Z}_2^{O(5)}$, $[1920, 240997]$:

$$
\begin{pmatrix}
0 & 0 & 0 & 0 & 1 \\
0 & 0 & 0 & 1 & 0 \\
1 & 0 & 0 & 0 & 0 \\
0 & 0 & 1 & 0 & 0 \\
0 & 1 & 0 & 0 & 0
\end{pmatrix},
\begin{pmatrix}
0 & -1 & 0 & 0 & 0 \\
0 & 0 & 0 & 1 & 0 \\
1 & 0 & 0 & 0 & 0 \\
0 & 0 & 0 & 0 & -1 \\
0 & 0 & 1 & 0 & 0
\end{pmatrix},
\begin{pmatrix}
1 & 0 & 0 & 0 & 0 \\
0 & -1 & 0 & 0 & 0 \\
0 & 0 & 1 & 0 & 0 \\
0 & 0 & 0 & 1 & 0 \\
0 & 0 & 0 & 0 & 1
\end{pmatrix},
\tag{B.12}
$$

$$
M_5(z) = \frac{z^{16} - z^{12} + z^8 - z^4 + 1}{(1-z^2)(1-z^4)^2(1-z^6)(1-z^{10})} .
\tag{B.13}
$$

5. $(((\mathbb{Z}_2)^4 : A_5) : \mathbb{Z}_4) \times \mathbb{Z}_2^{O(5)}$, $[640, 21536]$:

$$
\begin{pmatrix}
0 & 0 & 1 & 0 & 0 \\
1 & 0 & 0 & 0 & 0 \\
0 & 0 & 0 & 1 & 0 \\
0 & 1 & 0 & 0 & 0 \\
0 & 0 & 0 & 0 & 1
\end{pmatrix},
\begin{pmatrix}
0 & 0 & -1 & 0 & 0 \\
0 & 0 & 0 & 0 & 1 \\
1 & 0 & 0 & 0 & 0 \\
0 & 0 & 0 & -1 & 0 \\
0 & 1 & 0 & 0 & 0
\end{pmatrix},
\begin{pmatrix}
1 & 0 & 0 & 0 & 0 \\
0 & 1 & 0 & 0 & 0 \\
0 & 0 & -1 & 0 & 0 \\
0 & 0 & 0 & 1 & 0 \\
0 & 0 & 0 & 0 & 1
\end{pmatrix},
\tag{B.14}
$$

$$
M_5(z) = \frac{z^{12} + z^6 - z^2 + 1}{(1-z^2)^2(1-z^4)(1-z^8)(1-z^{10})} .
\tag{B.15}
$$

6. $((\mathbb{Z}_2)^4 : A_5)$, $[960, 11358]$:

$$
\begin{pmatrix}
0 & 1 & 0 & 0 & 0 \\
0 & 0 & -1 & 0 & 0 \\
0 & 0 & 0 & 0 & -1 \\
1 & 0 & 0 & 0 & 0 \\
0 & 0 & 0 & 1 & 0
\end{pmatrix},
\begin{pmatrix}
0 & 0 & 0 & 0 & 1 \\
0 & 0 & 1 & 0 & 0 \\
0 & 0 & 0 & -1 & 0 \\
-1 & 0 & 0 & 0 & 0 \\
0 & 1 & 0 & 0 & 0
\end{pmatrix},
\tag{B.16}
$$

$$
M_5(z) = \frac{z^{16} - z^{12} + z^8 - z^4 + 1}{(1-z^2)(1-z^4)^2(1-z^5)(1-z^6)} .
\tag{B.17}
$$

7. $((\mathbb{Z}_2)^4 : A_5) : \mathbb{Z}_4 - A$, $[320, 1635]$:

$$
\begin{pmatrix}
0 & 1 & 0 & 0 & 0 \\
0 & 0 & 0 & 0 & 1 \\
0 & 0 & -1 & 0 & 0 \\
-1 & 0 & 0 & 0 & 0 \\
0 & 0 & 0 & 1 & 0
\end{pmatrix},
\begin{pmatrix}
0 & 0 & 1 & 0 & 0 \\
-1 & 0 & 0 & 0 & 0 \\
0 & 0 & 0 & 0 & -1 \\
0 & -1 & 0 & 0 & 0 \\
0 & 0 & 0 & -1 & 0
\end{pmatrix},
\tag{B.18}
$$

$$
M_5(z) = \frac{z^{12} + z^6 - z^2 + 1}{(1-z^2)^2(1-z^4)(1-z^5)(1-z^8)} .
\tag{B.19}
$$

8. $((\mathbb{Z}_2)^4 : A_5) : \mathbb{Z}_4 - B, [320, 1635]$ :

$$\begin{pmatrix} 0 & 0 & 0 & 1 & 0 \\ 1 & 0 & 0 & 0 & 0 \\ 0 & 0 & 0 & 0 & 1 \\ 0 & 0 & 1 & 0 & 0 \\ 0 & 1 & 0 & 0 & 0 \end{pmatrix} \begin{pmatrix} 0 & 0 & -1 & 0 & 0 \\ 1 & 0 & 0 & 0 & 0 \\ 0 & 0 & 0 & 1 & 0 \\ 0 & -1 & 0 & 0 & 0 \\ 0 & 0 & 0 & 0 & -1 \end{pmatrix}, \tag{B.20}$$

$$M_5(z) = \frac{z^{16} - z^{14} - z^{11} + z^{10} + z^9 + z^7 + z^6 - z^5 - z^2 + 1}{(1 - z^2)^2 (1 - z^4)(1 - z^5)(1 - z^8)}. \tag{B.21}$$

9. $(((\mathbb{Z}_2)^4 : \mathbb{Z}_5) : \mathbb{Z}_2) \times \mathbb{Z}_2^{O(5)}, [320, 1636]$ :

$$\begin{pmatrix} 0 & 0 & 0 & 0 & -1 \\ 0 & 0 & -1 & 0 & 0 \\ -1 & 0 & 0 & 0 & 0 \\ 0 & -1 & 0 & 0 & 0 \\ 0 & 0 & 0 & -1 & 0 \end{pmatrix}, \begin{pmatrix} 0 & 0 & 0 & -1 & 0 \\ 0 & 0 & 1 & 0 & 0 \\ 0 & 1 & 0 & 0 & 0 \\ 1 & 0 & 0 & 0 & 0 \\ 0 & 0 & 0 & 0 & -1 \end{pmatrix}, \tag{B.22}$$

$$M_5(z) = \frac{z^{12} - z^{10} + 2z^6 - z^2 + 1}{(1 - z^2)^2 (1 - z^4)^2 (1 - z^{10})}. \tag{B.23}$$

10. $((\mathbb{Z}_2)^4 : \mathbb{Z}_5) : \mathbb{Z}_2 - A, [160, 234]$ :

$$\begin{pmatrix} 0 & 0 & 0 & -1 & 0 \\ 0 & 0 & -1 & 0 & 0 \\ 0 & -1 & 0 & 0 & 0 \\ 1 & 0 & 0 & 0 & 0 \\ 0 & 0 & 0 & 0 & -1 \end{pmatrix}, \begin{pmatrix} 0 & 0 & 0 & 1 & 0 \\ 1 & 0 & 0 & 0 & 0 \\ 0 & 0 & 0 & 0 & 1 \\ 0 & 0 & -1 & 0 & 0 \\ 0 & -1 & 0 & 0 & 0 \end{pmatrix}, \tag{B.24}$$

$$M_5(z) = \frac{z^{12} - z^{10} + 2z^6 - z^2 + 1}{(1 - z^2)^2 (1 - z^4)^2 (1 - z^5)}. \tag{B.25}$$

11. $((\mathbb{Z}_2)^4 : \mathbb{Z}_5) : \mathbb{Z}_2 - B, [160, 234]$ :

$$\begin{pmatrix} 0 & 0 & 0 & -1 & 0 \\ 0 & 0 & -1 & 0 & 0 \\ 0 & -1 & 0 & 0 & 0 \\ 1 & 0 & 0 & 0 & 0 \\ 0 & 0 & 0 & 0 & 1 \end{pmatrix}, \begin{pmatrix} 0 & 0 & 0 & -1 & 0 \\ 1 & 0 & 0 & 0 & 0 \\ 0 & 0 & 0 & 0 & 1 \\ 0 & 0 & -1 & 0 & 0 \\ 0 & 1 & 0 & 0 & 0 \end{pmatrix}, \tag{B.26}$$

$$M_5(z) = \frac{z^7 + 2z^6 - z^5 - z^2 + 1}{(1 - z^2)^2 (1 - z^4)^2 (1 - z^5)}. \tag{B.27}$$

12. $((\mathbb{Z}_2)^4 : \mathbb{Z}_5) \times \mathbb{Z}_2^{O(5)}, [160, 235]$ :

$$\begin{pmatrix} 1 & 0 & 0 & 0 & 0 \\ 0 & 1 & 0 & 0 & 0 \\ 0 & 0 & 1 & 0 & 0 \\ 0 & 0 & 0 & -1 & 0 \\ 0 & 0 & 0 & 0 & -1 \end{pmatrix}, \begin{pmatrix} 0 & 0 & 0 & -1 & 0 \\ -1 & 0 & 0 & 0 & 0 \\ 0 & 0 & 0 & 0 & -1 \\ 0 & 0 & -1 & 0 & 0 \\ 0 & -1 & 0 & 0 & 0 \end{pmatrix}, \tag{B.28}$$

$$M_5(z) = \frac{z^8 - 3z^6 + 5z^4 - 3z^2 + 1}{(1 - z^2)^4 (1 - z^{10})}. \tag{B.29}$$

13. $(\mathbb{Z}_2)^4 : \mathbb{Z}_5$, $[80,49]$:

$$
\begin{pmatrix}
1 & 0 & 0 & 0 & 0 \\
0 & 1 & 0 & 0 & 0 \\
0 & 0 & 1 & 0 & 0 \\
0 & 0 & 0 & -1 & 0 \\
0 & 0 & 0 & 0 & -1
\end{pmatrix},
\begin{pmatrix}
0 & 0 & 0 & 0 & 1 \\
1 & 0 & 0 & 0 & 0 \\
0 & 1 & 0 & 0 & 0 \\
0 & 0 & -1 & 0 & 0 \\
0 & 0 & 0 & -1 & 0
\end{pmatrix},
\tag{B.30}
$$

$$
M_5(z) = \frac{z^8 - 3z^6 + 5z^4 - 3z^2 + 1}{(1-z^2)^4(1-z^5)}.
\tag{B.31}
$$

14. $S_6 \times \mathbb{Z}_2^{O(5)}$, $[1440, 5842]$:

$$
\begin{pmatrix}
0 & 1 & 1 & 0 & 0 \\
-1 & 1 & 1 & -1 & 0 \\
0 & 0 & 0 & 1 & 0 \\
-1 & 0 & 1 & 0 & 0 \\
0 & 1 & 0 & -1 & -1
\end{pmatrix},
\begin{pmatrix}
0 & -1 & 0 & 0 & 0 \\
0 & -1 & 0 & 1 & 1 \\
0 & 0 & 0 & -1 & 0 \\
0 & 0 & 1 & 0 & 1 \\
1 & -1 & -1 & 1 & 0
\end{pmatrix},
\tag{B.32}
$$

$$
M_5(z) = \frac{z^8 + 1}{(1-z^2)(1-z^4)(1-z^6)^2(1-z^{10})}.
\tag{B.33}
$$

15. $S_6-A$, $[720, 763]$:

$$
\begin{pmatrix}
0 & -1 & 0 & 0 & 0 \\
0 & -1 & 0 & 1 & 1 \\
0 & 0 & 0 & -1 & 0 \\
0 & 0 & 1 & 0 & 1 \\
1 & -1 & -1 & 1 & 0
\end{pmatrix},
\begin{pmatrix}
0 & -1 & -1 & 0 & 0 \\
1 & -1 & -1 & 1 & 0 \\
0 & 0 & 0 & -1 & 0 \\
1 & 0 & -1 & 0 & 0 \\
0 & -1 & 0 & 1 & 1
\end{pmatrix},
\tag{B.34}
$$

$$
M_5(z) = \frac{1}{(1-z^2)(1-z^3)(1-z^4)(1-z^5)(1-z^6)}.
\tag{B.35}
$$

This irrep of $S_6$ is sometimes called the "standard" irrep.

16. $S_6-B$, $[720, 763]$:

$$
\begin{pmatrix}
0 & 1 & 1 & 0 & 0 \\
-1 & 1 & 1 & -1 & 0 \\
0 & 0 & 0 & 1 & 0 \\
-1 & 0 & 1 & 0 & 0 \\
0 & 1 & 0 & -1 & -1
\end{pmatrix},
\begin{pmatrix}
0 & 1 & 0 & 0 & 0 \\
0 & 1 & 0 & -1 & -1 \\
0 & 0 & 0 & 1 & 0 \\
0 & 0 & -1 & 0 & -1 \\
-1 & 1 & 1 & -1 & 0
\end{pmatrix},
\tag{B.36}
$$

$$
M_5(z) = \frac{-z^{17} + z^{15} + z^{14} - z^{11} - z^{10} - z^9 + z^8 + z^7 + z^6 - z^3 - z^2 + 1}{(1-z^2)^2(1-z^3)(1-z^4)(1-z^5)(1-z^6)}.
\tag{B.37}
$$

17. $A_6 \times \mathbb{Z}_2^{O(5)}$, $[720, 766]$:

$$
\begin{pmatrix}
0 & -1 & 0 & 1 & 1 \\
-1 & 1 & 1 & 0 & 0 \\
0 & -1 & 0 & 0 & 1 \\
-1 & 0 & 1 & 0 & 0 \\
0 & 1 & 1 & 0 & 0
\end{pmatrix},
\begin{pmatrix}
0 & -1 & -1 & 1 & 0 \\
1 & 0 & -1 & 0 & -1 \\
-1 & 0 & 0 & 0 & 0 \\
0 & 1 & 0 & 0 & -1 \\
1 & 0 & -1 & 0 & 0
\end{pmatrix},
\tag{B.38}
$$

$$
M_5(z) = \frac{z^{20} + z^{18} + z^8 + 1}{(1-z^2)(1-z^4)(1-z^6)^2(1-z^{10})}.
\tag{B.39}
$$

18. $S_5 \times \mathbb{Z}_2^{O(5)}$, [240, 189]:

$$
\begin{pmatrix}
0 & 1 & 0 & -1 & 0 \\
0 & 0 & 1 & 0 & 1 \\
0 & 0 & -1 & 0 & 0 \\
-1 & 0 & 1 & 0 & 1 \\
0 & 1 & 1 & 0 & 0
\end{pmatrix},
\begin{pmatrix}
0 & 0 & -1 & 0 & 0 \\
0 & -1 & 0 & 1 & 1 \\
0 & 0 & -1 & 0 & -1 \\
1 & -1 & -1 & 1 & 0 \\
0 & -1 & 0 & 0 & 1
\end{pmatrix},
\tag{B.40}
$$

$$
M_5(z) = \frac{z^{20} + z^{18} + z^{16} + 2z^{14} + 2z^{12} + z^{10} + 2z^8 + z^6 + 1}{(1-z^2)(1-z^4)(1-z^6)^2(1-z^{10})}.
\tag{B.41}
$$

19. $A_6$, [360, 118]:

$$
\begin{pmatrix}
0 & 0 & -1 & 0 & 0 \\
0 & 1 & 0 & 0 & -1 \\
0 & 0 & 0 & 0 & 1 \\
-1 & 1 & 1 & 0 & 0 \\
0 & 1 & 0 & -1 & -1
\end{pmatrix},
\begin{pmatrix}
0 & 1 & 1 & -1 & 0 \\
1 & -1 & -1 & 0 & 0 \\
0 & 1 & 1 & 0 & 0 \\
1 & 0 & -1 & 0 & -1 \\
0 & -1 & 0 & 0 & 0
\end{pmatrix},
\tag{B.42}
$$

$$
M_5(z) = \frac{z^{12} - z^9 + z^6 - z^3 + 1}{(1-z^2)(1-z^3)^2(1-z^4)(1-z^5)}.
\tag{B.43}
$$

20. $S_5-A$, [120, 34]:

$$
\begin{pmatrix}
0 & 0 & 0 & 1 & 0 \\
0 & 0 & -1 & 0 & 0 \\
0 & -1 & 0 & 1 & 0 \\
-1 & 0 & 0 & 0 & 0 \\
0 & 0 & 0 & 0 & 1
\end{pmatrix},
\begin{pmatrix}
0 & 1 & 0 & 0 & -1 \\
1 & -1 & -1 & 1 & 0 \\
0 & 1 & 0 & -1 & -1 \\
1 & 0 & -1 & 0 & 0 \\
0 & -1 & -1 & 1 & 0
\end{pmatrix},
\tag{B.44}
$$

$$
M_5(z) = \frac{z^{12} + z^{10} + z^9 + z^8 + z^6 + 1}{(1-z^2)(1-z^3)(1-z^4)(1-z^5)(1-z^6)}.
\tag{B.45}
$$

21. $S_5-B$, [120,34]:

$$
\begin{pmatrix}
0 & 1 & 0 & 0 & 0 \\
-1 & 0 & 1 & 0 & 1 \\
1 & 0 & -1 & 0 & 0 \\
0 & -1 & 0 & 1 & 1 \\
-1 & 0 & 0 & 0 & 0
\end{pmatrix},
\begin{pmatrix}
0 & -1 & 0 & 0 & 1 \\
-1 & 1 & 1 & -1 & 0 \\
0 & -1 & 0 & 1 & 1 \\
-1 & 0 & 1 & 0 & 0 \\
0 & 1 & 1 & -1 & 0
\end{pmatrix},
\tag{B.46}
$$

$$
M_5(z) = \frac{z^{14} - z^{13} + z^{12} - z^{11} + z^{10} - z^9 + 2z^8 - z^7 + 2z^6 - z^5 + z^4 - z^3 + z^2 - z + 1}{(1-z)(1-z^3)(1-z^4)(1-z^5)(1-z^6)}.
$$
$$
\tag{B.47}
$$

22. $A_5 \times \mathbb{Z}_2^{O(5)}$, [120, 35]:

$$
\begin{pmatrix}
-1 & 1 & 1 & -1 & 0 \\
0 & 0 & -1 & 0 & 0 \\
0 & 1 & 1 & -1 & 0 \\
0 & 0 & 0 & 0 & 1 \\
0 & -1 & -1 & 0 & 0
\end{pmatrix},
\begin{pmatrix}
0 & -1 & 0 & 1 & 0 \\
1 & -1 & -1 & 0 & 0 \\
0 & 0 & 0 & 1 & 0 \\
1 & 0 & -1 & 0 & -1 \\
0 & -1 & -1 & 0 & 0
\end{pmatrix},
\begin{pmatrix}
0 & 0 & -1 & 0 & 0 \\
1 & -1 & -1 & 0 & 0 \\
-1 & 0 & 0 & 0 & 0 \\
0 & -1 & 0 & 0 & 1 \\
1 & -1 & -1 & 1 & 0
\end{pmatrix},
$$
$$
\tag{B.48}
$$

$$
M_5(z) = \frac{2z^{18} + z^{14} + 3z^{12} + z^{10} + 2z^8 + 2z^6 + z^4 - z^2 + 1}{(1-z^2)^2(1-z^6)^2(1-z^{10})}.
\tag{B.49}
$$

23. $A_5$, [60,5]:

$$\begin{pmatrix} 0 & 1 & 1 & -1 & 0 \\ -1 & 0 & 1 & 0 & 1 \\ 1 & 0 & 0 & 0 & 0 \\ 0 & -1 & 0 & 0 & 1 \\ -1 & 0 & 1 & 0 & 0 \end{pmatrix}, \begin{pmatrix} 1 & -1 & -1 & 1 & 0 \\ 0 & 0 & 1 & 0 & 0 \\ 0 & -1 & -1 & 1 & 0 \\ 0 & 0 & 0 & 0 & -1 \\ 0 & 1 & 1 & 0 & 0 \end{pmatrix},$$ (B.50)

$$M_5(z) = \frac{z^{10} - z^8 + z^6 + z^5 + z^4 - z^2 + 1}{(1-z^2)^2(1-z^3)^2(1-z^5)}.$$ (B.51)

We now give the corresponding information about the Lie subgroups. The Hilbert series $H(z)$ provides in this case information analogous to the Molien series of finite groups.

- SO(5): The generators are given by the anti-symmetric matrices $(R_{ij})^{kl} = \delta_i^k \delta_j^l - \delta_i^l \delta_j^k$, with $i \neq j$. The Hilbert series is

$$H(z) = 1/(1-z^2).$$ (B.52)

- $SO(3)_T$ : The generators $J_1, J_2$ and $J_3$ of $SO(3)_T \subset SO(5)$ are given in (A.10). The Hilbert series is [49]

$$H(z) = \frac{1}{(1-z^2)(1-z^3)}.$$ (B.53)

- $SO(3)_T \times \mathbb{Z}_2^{O(5)}$ : The nontrivial $\mathbb{Z}_2^{O(5)}$ element acts as

$$\begin{pmatrix} -1 & 0 & 0 & 0 & 0 \\ 0 & -1 & 0 & 0 & 0 \\ 0 & 0 & -1 & 0 & 0 \\ 0 & 0 & 0 & -1 & 0 \\ 0 & 0 & 0 & 0 & -1 \end{pmatrix}.$$ (B.54)

The Hilbert series is

$$H(z) = \frac{1}{1 - z^2 - z^6 + z^8}.$$ (B.55)

# C  GAP basics

In this appendix, we collect some basic usage of GAP. To specify the finite group that one wish to work with, one can either use the Small Group Id (if known) as explained in (B.1), or use the built-in commands if the group has a commonly known name, such as

```
g:=SymmetricGroup(5);
```
(C.1)

for the symmetric group $S_5$ and

```
g:=AlternatingGroup(5);
```
(C.2)

for the alternating group $A_5$. To find the generators of the group, one can use

```
GeneratorsOfGroup(g);
```
(C.3)

The resulting set is expressed in terms of abstract group elements; in the case of groups consisting of permutations, the set will be a list of permutations written in cycle notations.

For two groups $\mathfrak{G}_1$ and $\mathfrak{G}_2$, one can check whether they are isomorphic, as explained in (21).

To calculate and display the character table, one should use the command in (9). One can also calculate the (second) Frobenius–Schur indicator the irreps as in (10).

To figure out a matrix representation of a irreps, one need to use (B.4) and (B.5). Before that, one need to extract characters of the irreps from the character table, as in (B.3). One can also easily find the Molien function of an irrep using (24).

For a group $\mathfrak{G}$, one can find all its maximal subgroups, up to conjugation, as was explained in (18).

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
