# Peer review of "Classifying irreducible fixed points of five scalar fields in perturbation theory"

_SciPost Physics, doi:SciPost Phys. 16, 040 (2024)_

## Round 2 · Referee Report · Anonymous · 2023-9-19

Strengths
1- Clearly written and pedagogical paper
Weaknesses
1- Results are too limited and narrow in scope
Report
This work explores the space of fixed points of theories with 5 scalar fields in $d=4-\varepsilon$ dimensions. It follows the largely standard practice of imposing a global symmetry from the beginning and looking for fixed points that may exist with that global symmetry. The global symmetry imposed is any "minimal I&L" subgroup of $O(5)$ in the authors' terminology. While some complex solutions to the beta function equations are discussed, this work focuses on real solutions corresponding to unitary fixed points.
The results presented in this paper do not raise any suspicions with regards to their validity. My main criticism is that they are so limited and narrow in scope that make the paper appear incomplete. In similar situations the lack of positive results would be followed by a relaxation of some of the assumptions involved in an attempt to go beyond what is already known. Such an attempt is not pursued by the authors, who instead resort to writing a pedagogical paper that does not end up containing a sufficient amount of original research to justify publication.
I would recommend that the authors attempt to relax some of the imposed conditions in order to potentially go beyond what is already known, or at least describe in some detail how the space of fixed points might be enlarged in that case. For example, going from 1 to 2 mass terms, thus relaxing the condition of irreducibility, could already allow for a richer yet manageable set of solutions. As it stands, I cannot recommend publication of this work as a research paper.
Author: Junchen Rong on 2023-12-06 [id 4172]
(in reply to Report 1 on 2023-09-19)We thank the referee for having taken time to read our paper and give suggestions.
We want to emphasize the fact that there is a clear distinction between our work and previous results. Our paper is the first one to undertake the SYSTEMATIC classification of irreducible fixed points with five scalars in 4-epsilon expansion, while previous work approaching these fixed points proceeded on a case-by-case and ad hoc basis. Even though all the fixed points we found have been encountered before, this was not at all a forgone conclusion but a result of meticulous and nontrivial calculations. Our paper thus finishes the N=5 classification problem for the first time. It’s not correct to say that our paper has no new results, as the very proof that there are no other fixed points of this type is a new result which is obtained here for the first time. There are many famous no-go theorems in physics which rigorously delineate the bounds of possible; ours is a result of this type.
To reach this conclusion, we had to work out some auxiliary results which are also new, e.g. for the first time work out all the irreducible subgroups of O(5), and their group-subgroup relations. This result is presented in Fig. 2 and Fig 3. To our knowledge, such a table was not known before even in math literature. Another way of presenting our result would be to say that we have thoroughly analyzed the Laudau theory of all these groups, and some of them have not been considered in the literature before. Too bad this did not lead to new fixed points, but this could not have been guessed beforehand. While our method dates back to the work of Toledano, Michel, Toledano, and Brezin in 1985, this method is thoroughly forgotten and has not been used in the recent works on fixed points in the 4-epsilons expansion. In addition to reviving the method, we added many new features to it. This will be an indispensable help to anyone who dares to go further in the classification problem than we did.
Relaxing the one mass condition, as suggested by the referee may be an interesting future project (although such fixed points are physically less interesting). It is certain to require even longer computations than the ones we undertook, as the number of subgroups to consider will be much higher. It cannot be achieved by a small addition to the paper but should be postponed to a future publication, which however may be undertaken only once our first work is published.

---

## Round 2 · Referee Report · Edoardo Lauria · 2023-12-26

Strengths
1. Pedagogical and very well written
2. Very systematic search
3. Rigorous results
Report
The authors classify all unitary and RG-stable fixed points with $N=5$ scalars in $d=4-\varepsilon$ dimensions, with irreducible symmetry group $G\in O(N)$ and satisfying the Landau condition ($N=5$ ILS fixed points). The irreducibility condition (I) is the requirement that there is a single mass term compatible with $G$, so that the fixed point (if it exists) can be reached by tuning a single relevant parameter. The Landau condition (L) is the requirement that there is no cubic term compatible with $G$, and this is important because transitions that violate it are generically expected to be of first order, at least for $d>2$. Unitarity and RG-stability of the IL fixed points are investigated at one loop, using the beta functions for generic scalar theories with quartic interactions that are known from the literature.
The result of this classification is that the only ILS fixed points with $N=5$ are: the Wilson-Fisher fixed point (with $O(5)$ symmetry) and the cubic fixed point (with $(\mathbb{Z}_2)^5 \rtimes S_5$ symmetry). This important result follows from a very rigorous and systematic search, which is sharpened by group-theoretic arguments (conditions I&L) and made efficient thanks to the help of a computer program called GAP.
The presentation of the paper is clear and pedagogic. The authors explain in detail each step of their algorithm, how to implement them in GAP, what are the subtleties if one wants to relax some of their assumptions. For anyone interested in similar classification problems, thanks to this paper, I think the path is now clear.
The scientific quality of this work is top level. There is no doubt that it contains new important results, and that such results are correct. In particular, the efficient algorithm developed by the authors allowed them to solve a problem that was open since the work from Toledano et al. in 1985, who carried out a similar classification for ILS fixed points with $N=4$.
In summary, this paper meets all the general acceptance criteria of SciPost, as well as some of the expectations (2&3). I am therefore very happy to recommend this paper for publication.

---

## Round 2 · Referee Report · Anonymous · 2023-12-28

Report
This works classifies Wilson-Fisher type fixed points in the theory of five real scalar fields.
The authors consider subgroups G of the maximal global symmetry group O(5) and then investigate whether there exist fixed points which are stable in the space of G-invariant RG flows, in the sense that only a single relevant coupling needs to be tuned to reach them.
This stability demand is then further split into the absence of non-trivial quadratic invariants, of cubic invariants, and of relevant quartic operators at the fixed points. The authors call these the I, L and S conditions. These conditions in particular severely constrain the possible subgroups G.
In more detail, the physics demands to consider subgroups G of O(5) equipped with a five-dimensional faithful representation $\rho$ (or rather its isomorphism class), up to outer automorphisms. In the language of the authors these are matrix groups (cf. their Lemma 3.1). They outline an algorithm to obtain all the matrix subgroups of O(5) thay obey the I and L conditions using GAP. With this list in hand the authors consider the one-loop beta functions to verify the S condition of all the real perturbative fixed points.
The end result is that the classification yields only two possibilities, the Cubic(5) and O(5) fixed points. Since these were both previously known, the main new result of the paper is the rigorous proof that no other ILS fixed points exist for N = 5.
The paper is well written and contains only minor typos. The logical steps are clear, although the authors could have opted for a more mathematical style to dispel any doubts about the rigor of their result.
Altogether the paper easily meets the threshold for publication in SciPost.

---

## Editorial Decision

published